# Controlling Counterfactual Harm in Decision Support Systems Based on Prediction Sets

**Eleni Straitouri**
Max Planck Institute for
Software Systems
Kaiserslautern, Germany
`estraitouri@mpi-sws.org`

**Suhas Thejaswi**
Max Planck Institute for
Software Systems
Kaiserslautern, Germany
`thejaswi@mpi-sws.org`

**Manuel Gomez Rodriguez**
Max Planck Institute for
Software Systems
Kaiserslautern, Germany
`manuelgr@mpi-sws.org`

## Abstract

Decision support systems based on prediction sets help humans solve multiclass classification tasks by narrowing down the set of potential label values to a subset of them, namely a prediction set, and asking them to always predict label values from the prediction sets. While this type of systems have been proven to be effective at improving the average accuracy of the predictions made by humans, by restricting human agency, they may cause harm—a human who has succeeded at predicting the ground-truth label of an instance on their own may have failed had they used these systems. In this paper, our goal is to control how frequently a decision support system based on prediction sets may cause harm, by design. To this end, we start by characterizing the above notion of harm using the theoretical framework of structural causal models. Then, we show that, under a natural, albeit unverifiable, monotonicity assumption, we can estimate how frequently a system may cause harm using only predictions made by humans on their own. Further, we also show that, under a weaker monotonicity assumption, which can be verified experimentally, we can bound how frequently a system may cause harm again using only predictions made by humans on their own. Building upon these assumptions, we introduce a computational framework to design decision support systems based on prediction sets that are guaranteed to cause harm less frequently than a user-specified value using conformal risk control. We validate our framework using real human predictions from two different human subject studies and show that, in decision support systems based on prediction sets, there is a trade-off between accuracy and counterfactual harm.

## 1 Introduction

The principle of "first, do no harm" holds profound significance in a variety of professions across multiple high-stakes domains. For example, in the field of medicine, doctors swear an oath to prioritize their patient's well-being or, in the legal justice system, preserving the innocence of individuals is paramount. As a result, in all of these domains, rules and guidelines have been established to prevent decision makers—doctors or judges—from making decisions that harm individuals—patients or suspects. In recent years, it has been increasingly argued that a similar principle should apply to decision support systems using machine learning algorithms in high-stakes domains [1–4].[1]

The definition of harm is not unequivocally agreed upon, however, the most widely accepted definition is the counterfactual comparative account of harm (in short, counterfactual harm) [5–7], which we adopt in our work. Under this definition, an action causes harm to an individual if they would have

---

[1]The European Unions' AI act mentions the term "harm" more than 35 times and points out that, its crucial role in the design of algorithmic systems must be defined carefully.

been in a worse state had the action been taken. Building upon this definition, we say that a decision support system causes harm to an individual if a decision maker would have made a worse decision about the individual had they used the system.

In machine learning for decision support, one of the main focus has been classification tasks. Here, the most studied setting assumes the decision support system uses a classifier to predict the value of a (ground-truth) label of interest and a human expert uses the predicted value to update their own prediction [8–12]. Unfortunately, in this setting, it is yet unclear how to guarantee that the (average) accuracy of the predictions made by an expert who uses the system is higher than the accuracy of the predictions made by the expert and the classifier on their own, what is often referred to as human-AI complementarity [13–16]. In this context, a recent line of work [17, 18] have argued, both theoretically and empirically, that an alternative setting may enable human-AI complementarity. In this alternative setting, the decision support system helps a human expert by providing a set of label predictions, namely a prediction set, and asking them to always predict a label value from the set. The key principle is that, by restricting human agency, good performance does not depend on the expert developing a good sense of when to predict a label from the prediction set. In this context, it is also worth noting that Google has recently developed a tool that uses patient history and skin condition images to provide decision support using prediction sets [19], and a study by Jain et al. [20] has found that physicians and nurses using this tool improved diagnoses for 1 in every 8 to 10 cases.

In this work, we argue that the same principle that enables human-AI complementarity on decision support systems based on prediction sets may also cause counterfactual harm—a human expert who has succeeded at predicting the label of an instance on their own may have failed had they used these systems. Consequently, our goal is to design decision support systems based on prediction sets that are guaranteed to cause, in average, less counterfactual harm than a user-specified value.

**Our contributions.** We start by formally characterizing the predictions made by a decision maker using a decision support system based on prediction sets using a structural causal model (SCM) and, based on this characterization, formalize our notion of counterfactual harm. In general, since counterfactual harm lies within level three in the "ladder of causation" [21], it is not (partially) identifiable—it cannot be estimated (bounded) from data. However, we show that, under a natural counterfactual monotonicity assumption on the predictions made by decision makers using decision support systems based on prediction sets, counterfactual harm is identifiable. Further, we show that, under a weaker interventional monotonicity assumption, which can be verified experimentally, the average counterfactual harm is partially identifiable. Then, building upon these assumptions, we develop a computational framework to design decision support systems based on prediction sets that are guaranteed to cause, in average, less counterfactual harm than a user-specified value using conformal risk control [22]. Finally, we validate our framework using real human predictions from two different human subject studies and show that, in decision support systems based on prediction sets, there is a trade-off between accuracy and counterfactual harm.

**Further related work.** Our work builds upon further related work on set-valued predictors, critiques of prediction optimization, counterfactual harm and algorithmic triage.

Set-valued predictors output a set of label values, namely a prediction set, rather than single labels [23]. However, set-valued predictors have not been typically designed nor evaluated by their ability to help human experts make more accurate predictions [24–29]. Only very recently, an emerging line of work has shown that conformal predictors, a specific type of set-valued predictors, may help human experts make more accurate predictions [17, 18, 30–33]. Within this line of work, the work most closely related to ours is by Straitouri et al. [17, 18], which has introduced the setting, and counterfactual and interventional monotonicity properties we build upon.

Prediction optimization has been recently put into question in the context of decision support [34–36]. More specifically, it has been argued that optimizing decision support systems to improve prediction accuracy does not always translate to better decision-making. Our work aligns with this critique since we argue that improving prediction accuracy may come at the cost of counterfactual harm.

The literature on counterfactual harm in machine learning is still quite small and has focused on traditional machine learning settings in which machine learning models replace human decision makers and make automated decisions [1–4]. Within this literature, the work most closely related to ours is by Richens et al. [1], which also uses a structural causal model to define counterfactual harm. However, their definition of counterfactual harm differs in a subtle, but important, way from ours. In

our setting, their definition of counterfactual harm would compare the accuracy of a factual prediction made by an expert using the decision support system against the counterfactual prediction that the same expert would have made on their own. That means, under their definition, one would need to deploy the system to estimate the harm it may cause, potentially causing harm. In contrast, under our definition, one does not need to deploy the system to estimate (or bound) the harm it may cause, as it will become clear in Section 4, and thus we argue that our definition may be more practical.

Learning under algorithmic triage seeks to develop classifiers that make predictions for a given fraction of the samples and leave the remaining ones to human experts, as instructed by a triage policy [37–42]. In contrast, in our work, for each sample, a classifier is used to construct a prediction set and a human expert needs to predict a label value from the set. In this context, it is also worth noting that learning under algorithmic triage has been extended to reinforcement learning settings [43–46].

## 2 Decision support systems based on prediction sets

We consider a multiclass classification task in which, for each task instance, a human expert has to predict the value of a ground-truth label $y \in \mathcal{Y} = \{1, \ldots, L\}$. Then, our goal is to design a decision support system $\mathcal{C} : \mathcal{X} \to 2^{\mathcal{Y}}$ that, given a set of features $x \in \mathcal{X}$, helps the expert by narrowing down the label values to a subset of them $\mathcal{C}(x) \subseteq \mathcal{Y}$, namely a prediction set. Here, we focus on a setting in which the system asks the expert to *always* predict a label value $\hat{y}$ from the prediction set $\mathcal{C}(x)$. Note that, by restricting the expert's agency, good performance does not depend on the human expert developing a good sense of when to predict a label from the prediction set. Moreover, we assume that the set of features, the ground-truth label and the expert's prediction are sampled from an unknown fixed distribution[2], *i.e.*, $x, y \sim P(X, Y)$ and $\hat{y} \sim P(\hat{Y} \mid X, Y, \mathcal{C}(X))$.

Further, similarly as in Straitouri et al. [17, 18], we consider that, given a set of features $x \in \mathcal{X}$, the system constructs the prediction set $\mathcal{C}(x)$ using the following set-valued predictor [23]. First, the set-valued predictor ranks each potential label value $y \in \mathcal{Y}$ using the softmax output of a pre-trained classifier $m_y(x) \in [0, 1]$. Then, given a user-specified threshold $\lambda \in [0, 1]$, it uses the resulting ranking to construct the prediction set $\mathcal{C}(x) = \mathcal{C}_\lambda(x)$ as follows:

$$\mathcal{C}_\lambda(x) = \{y_{(i)}\}_{i=1}^k, \text{ with } k = 1 + \sum_{j=2}^{L} \mathbb{1}\left\{m_{y_{(j)}}(x) \geq 1 - \lambda\right\}, \tag{1}$$

where $\cdot_{(i)}$ denotes the $i$-th label value in the ranking. Here, note that, for $\lambda = 0$, the prediction set contains just the top ranked label value, for $\lambda = 1$, it contains all label values.[3]

Given the above parameterization, one may just focus on finding the optimal threshold $\lambda^*$ under which the human expert achieves the highest average accuracy as in Straitouri et al. [17, 18], *i.e.*,

$$\lambda^* = \underset{\lambda \in [0,1]}{\operatorname{argmax}} A(\lambda), \text{ with } A(\lambda) = \mathbb{E}_{X,Y \sim P(X,Y), \hat{Y} \sim P(\hat{Y} \mid X, Y, \mathcal{C}_\lambda(X))}[\mathbb{1}\{\hat{Y} = Y\}]. \tag{2}$$

However, this focus does not prevent the resulting system $\mathcal{C}_\lambda$ from causing harm—an expert may succeed to predict the value of the ground-truth label on their own on instances in which they would have failed had they used $\mathcal{C}_\lambda$. In this work, our goal is to design a computational framework that, given a user-specified bound $\alpha \in [0, 1]$, finds the set of $\lambda$ values which are all guaranteed to cause less harm, in average, than the bound $\alpha$.

**Remark.** We would like to clarify that, if one sets the value of the threshold $\lambda$ to be roughly the $1 - \alpha$ quantile of the empirical distribution of the scores $1 - m_y(x)$ in a calibration set, then, the set valued predictor defined by Eq. 1 is equivalent to a vanilla conformal predictor with nonconformity scores $1 - m_y(x)$ and coverage $1 - \alpha$. Under this view, it becomes apparent that, by searching for $\lambda$ values in $[0, 1]$ that are harm controlling, we are essentially searching for vanilla conformal predictors that are harm controlling. In this context, we would like to further clarify that our framework is agnostic to the choice of nonconformity score or more generally, the set-valued predictor, used to construct the prediction sets [47–49]. Motivated by this observation, in Appendix E, we include additional experiments where we evaluate our framework using a more complex set-valued predictor [49].

---

[2]We denote random variables with capital letters and realizations of random variables with lowercase letters.
[3]The assumption that $m_y(x) \in [0, 1]$ and $\lambda \in [0, 1]$ is without loss of generality.

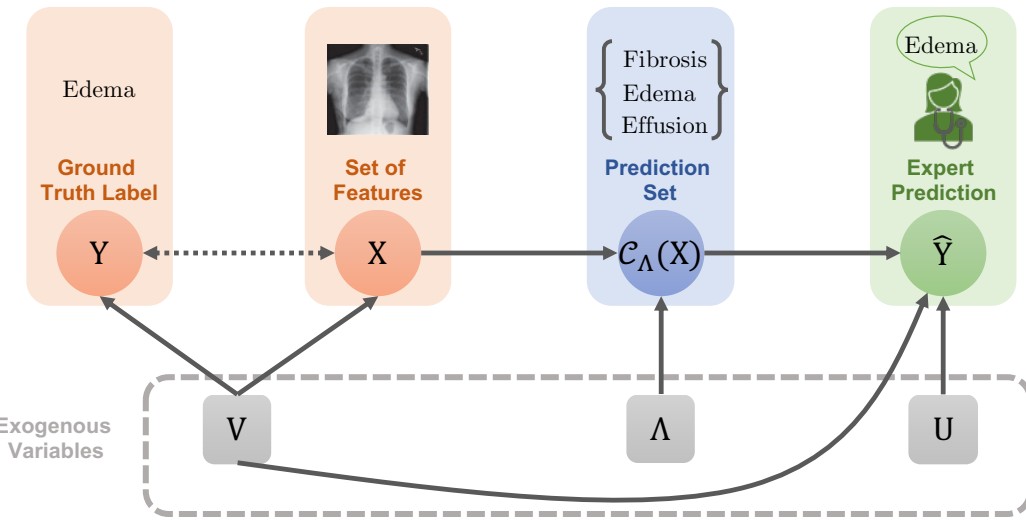

Figure 1: Our structural causal model $\mathcal{M}$. Circles represent endogenous random variables and boxes represent exogenous random variables. The value of each endogenous variable is given by a function of the values of its ancestors, as defined by Eq. 3. The value of each exogenous variable is sampled independently from a given distribution.

## 3 Counterfactual harm of decision support systems

To formalize our notion of harm, we characterize how human experts make predictions using a decision support system $\mathcal{C}$ via a structural causal model (SCM) [21], which we denote as $\mathcal{M}$. More specifically, similarly as in Straitouri et al. [18], we define $\mathcal{M}$ by the following set of assignments:

$$X = f_X(V), \qquad Y = f_Y(V), \qquad \mathcal{C}_\Lambda(X) = f_\mathcal{C}(\Lambda, X), \qquad \hat{Y} = f_{\hat{Y}}(U, V, \mathcal{C}_\Lambda(X)), \qquad (3)$$

where $\Lambda$, $U$ and $V$ are exogenous random variables and $f_X$, $f_Y$, $f_\mathcal{C}$ and $f_{\hat{Y}}$ are given functions.[4] The exogenous variables $\Lambda$, $U$ and $V$ characterize the user-specified threshold, the expert's individual characteristics and the data generating process, respectively. The function $f_\mathcal{C}$ is directly defined by Eq. 1, *i.e.*, $f_\mathcal{C}(\Lambda, X) = \mathcal{C}_\Lambda(X)$. Further, as argued elsewhere [21], we can always find a distribution for the exogenous variables $\Lambda$, $U$ and $V$ as well as a functional form for the functions $f_X$, $f_Y$ and $f_{\hat{Y}}$ such that the distributions of the features, the ground-truth label and the expert's prediction introduced in Section 2 are given by the observational distribution entailed by the SCM $\mathcal{M}$. For ease of exposition, we assume that, under no interventions, the distribution of the exogenous variable $\Lambda$ is $P(\Lambda) = \mathbb{1}\{\Lambda = 1\}$ and thus human experts make predictions on their own. Figure 1 shows a visual representation of our SCM $\mathcal{M}$.

Building upon the above characterization, we are now ready to formalize the following notion of counterfactual harm, which essentially compares the accuracy of a factual prediction made by an expert on their own against the counterfactual prediction that the same expert would have made had they used a decision support system $\mathcal{C}_\lambda$:

**Definition 1 (Counterfactual Harm)** *For any $x, y, \hat{y} \sim P^\mathcal{M}$, the counterfactual harm that a decision support system $\mathcal{C}_\lambda$ would have caused, if deployed, is given by[5]*

$$h_\lambda(x, y, \hat{y}) = \mathbb{E}_{\hat{Y} \sim P^{\mathcal{M};do(\Lambda=\lambda) \mid \hat{Y}=\hat{y}, X=x, Y=y}(\hat{Y})}[\max\{0, \mathbb{1}\{\hat{y} = y\} - \mathbb{1}\{\hat{Y} = y\}\}], \qquad (4)$$

*where $do(\Lambda = \lambda)$ denotes a (hard) intervention on the exogenous variable $\Lambda$.*

Here, note that counterfactual harm can only be nonzero if the expert has made a successful prediction on their own, *i.e.*, $\hat{y} = y$. Otherwise, the expert's prediction $\hat{y}$ could not have become worse had they used the decision support system $\mathcal{C}_\lambda$.

---

[4]The functions $f_X$, $f_Y$, $f_\mathcal{C}$ and $f_{\hat{Y}}$ are causal mechanisms and not equations that can be manipulated [50].

[5]To denote interventions in a counterfactual distribution, we follow the notation by Peters et al. [51]. Refer to Appendix A for a comparison to Pearl's notation [50].

Given the above definition of counterfactual harm and a user-specified bound $\alpha \in [0, 1]$, our goal is to find the largest harm-controlling set of values $\Lambda(\alpha) \subseteq [0, 1]$ such that, for each $\lambda \in \Lambda(\alpha)$, it holds that the counterfactual harm is, in expectation across all possible instances, smaller than $\alpha$, *i.e.*,

$$\Lambda(\alpha) = \left\{ \lambda \in [0, 1] \mid H(\lambda) = \mathbb{E}_{X,Y,\hat{Y} \sim P^{\mathcal{M}}(X,Y,\hat{Y})}[h_\lambda(X, Y, \hat{Y})] \leq \alpha \right\}. \tag{5}$$

At this point, we cannot expect to find the set $\Lambda(\alpha)$ because counterfactual harm lies within level three in the "ladder of causation" [21] and thus it is not identifiable from observational data without further assumptions. However, in what follows, we will show that, under certain assumptions, the average counterfactual harm $H(\lambda)$ is (partially) identifiable, *i.e.*, it can be estimated (bounded) using observational data.

**Comparison to Richens's definition of counterfactual harm.** Richens et al. [1] define counterfactual harm as follows:

$$h_\lambda(x, y, \hat{y}) = E_{\hat{Y} \sim P^{\mathcal{M} \mid \hat{Y}=\hat{y}, X=x, Y=y}(\hat{Y})}[\max\{0, \mathbb{1}\{\hat{Y} = y\} - \mathbb{1}\{\hat{y} = y\}\}],$$

where $x, y, \hat{y} \sim P^{\mathcal{M}; do(\Lambda=\lambda)}$ and $\lambda = 1$ is considered to be the *default action* in the language of Richens et al. [1]. This definition implicitly assumes that the system $\mathcal{C}_\lambda$ is deployed and it compares the factual prediction $\hat{y}$ made by an expert using the deployed system against the counterfactual prediction $\hat{Y}$ had the expert made on their own. On the contrary, our definition does not assume that the system $\mathcal{C}_\lambda$ is deployed and instead it compares the factual prediction $\hat{y}$ made by an expert on their own against the counterfactual prediction $\hat{Y}$ had the expert made using the system $\mathcal{C}_\lambda$.

# 4 Counterfactual harm under counterfactual and interventional monotonicity

In this section, we analyze counterfactual harm $h_\lambda(x, y, \hat{y})$, as defined in Eq. 4, from the perspective of causal identifiability under two natural monotonicity assumptions—counterfactual monotonicity and interventional monotonicity. Both of these assumptions, which were first studied by Straitouri et al. [18], formalize the intuition that increasing the number of label values in a prediction set increases its difficulty.

Under counterfactual monotonicity, for any $x \in \mathcal{X}$ and $\lambda, \lambda' \in [0, 1]$ such that $Y \in \mathcal{C}_\lambda(x) \subseteq \mathcal{C}_{\lambda'}(x)$, if an expert has succeeded at predicting $Y$ using $\mathcal{C}_{\lambda'}$, they would have also succeeded had they used $\mathcal{C}_\lambda$ and, conversely, if they have failed at predicting $Y$ using $\mathcal{C}_\lambda$, they would have also failed had they used $\mathcal{C}_{\lambda'}$, while holding "everything else fixed". More formally, counterfactual monotonicity is defined as follows:

**Assumption 1 (Counterfactual Monotonicity)** *Counterfactual monotonicity holds if and only if, for any $x \in \mathcal{X}$ and any $\lambda, \lambda' \in [0, 1]$ such that $Y \in \mathcal{C}_\lambda(x) \subseteq \mathcal{C}_{\lambda'}(x)$, we have that*

$$\mathbb{1}\{f_{\hat{Y}}(u, v, \mathcal{C}_\lambda(x)) = Y\} \geq \mathbb{1}\{f_{\hat{Y}}(u, v, \mathcal{C}_{\lambda'}(x)) = Y\} \tag{6}$$

*for any $u \sim P^{\mathcal{M}}(U)$ and $v \sim P^{\mathcal{M}}(V \mid X = x)$.*

Under interventional monotonicity, for any $x \in \mathcal{X}$ and $\lambda, \lambda' \in [0, 1]$ such that $Y \in \mathcal{C}_\lambda(x) \subseteq \mathcal{C}_{\lambda'}(x)$, the probability that experts succeed at predicting $Y$ using $\mathcal{C}_\lambda$ is equal or greater than using $\mathcal{C}_{\lambda'}$. More formally, interventional monotonicity is defined as follows[6]:

**Assumption 2 (Interventional Monotonicity)** *Interventional monotonicity holds if and only if, for any $x \in \mathcal{X}$, $y \in \mathcal{Y}$, and $\lambda, \lambda' \in [0, 1]$ such that $y \in \mathcal{C}_\lambda(x) \subseteq \mathcal{C}_{\lambda'}(x)$, we have that*

$$P^{\mathcal{M}; do(\Lambda=\lambda)}(\hat{Y} = Y \mid X = x, Y = y) \geq P^{\mathcal{M}; do(\Lambda=\lambda')}(\hat{Y} = Y \mid X = x, Y = y), \tag{7}$$

*where the probability is over the exogenous random variables $U$ and $V$ characterizing the expert's individual characteristics and the data generating process, respectively.*

In what follows, we first show that, under the counterfactual monotonicity assumption, counterfactual harm is identifiable (we provide all proofs in Appendix B):

---

[6]In Straitouri et al. [18], interventional monotonicity is originally defined unconditionally of the value of ground-truth label $Y$.

**Proposition 1** *Under the counterfactual monotonicity assumption, for any $x, y, \hat{y} \sim P^{\mathcal{M}}$, the counterfactual harm that a decision support system $\mathcal{C}_\lambda$ would have caused, if deployed, is given by*

$$h_\lambda(x, y, \hat{y}) = \mathbb{1}\{\hat{y} = y \wedge y \notin \mathcal{C}_\lambda(x)\}. \tag{8}$$

As an immediate consequence, we can conclude that, under counterfactual monotonicity, the average counterfactual harm $H(\lambda)$, as defined in Eq. 5, is identifiable and can be estimated using observational data sampled from $P^{\mathcal{M}}$. However, since the inequality condition of the counterfactual monotonicity assumption compares counterfactual predictions and thus cannot be experimentally verified, one should be cautious about using the above proposition to estimate counterfactual harm in high-stakes applications.

Next, we show that, under the interventional monotonicity assumption, the average counterfactual harm is partially identifiable:

**Proposition 2** *Under the interventional monotonicity assumption, the average counterfactual harm $H(\lambda)$ that a decision support system $\mathcal{C}_\lambda$ would have caused, if deployed, satisfies that*

$$\mathbb{E}_{X,Y,\hat{Y} \sim P^{\mathcal{M}}(X,Y,\hat{Y})}[\mathbb{1}\{\hat{Y} = Y \wedge Y \notin \mathcal{C}_\lambda(X)\}] \leq H(\lambda)$$
$$\leq \mathbb{E}_{X,Y,\hat{Y} \sim P^{\mathcal{M}}(X,Y,\hat{Y})}[\mathbb{1}\{\hat{Y} = Y \wedge Y \notin \mathcal{C}_\lambda(X)\} + \mathbb{1}\{\hat{Y} \neq Y \wedge Y \in \mathcal{C}_\lambda(X)\}]. \tag{9}$$

Importantly, note that, in the above proposition, the lower bound on the left hand side of Eq. 9 matches the average counterfactual harm under the counterfactual monotonicity assumption and thus, holding "everything else fixed", the average counterfactual harm under interventional monotonicity is always greater or equal than the average counterfactual harm under counterfactual monotonicity. Moreover, further note that the inequality condition of the interventional monotonicity assumption compares interventional probabilities and thus we can experimentally verify it (see Appendix F), lending support to using the above proposition to bound average counterfactual harm in high-stakes applications.

## 5 Controlling counterfactual harm using conformal risk control

In this section, we develop a computational framework that, given a decision support system $\mathcal{C}_\lambda$ and a user-specified bound $\alpha$, aims to find the largest harm-controlling set $\Lambda(\alpha)$, as defined in Eq. 5. In the development of our framework, we will first assume that counterfactual monotonicity holds and, later on, we will relax this assumption, and assume instead that interventional monotonicity holds.

Our framework builds upon the the idea of conformal risk control, which has been introduced very recently by Angelopoulos et al. [22]. Given any monotone loss function $\ell(\mathcal{C}_\lambda(X), Y)$ with respect to $\lambda$ and a calibration set $\{(X_i, Y_i)\}_{i=1}^n$, with $(X_i, Y_i) \sim P(X, Y)$, conformal risk control finds a value of $\lambda$ under which the expected loss of a test sample $(X_{n+1}, Y_{n+1}) \sim P(X, Y)$ does not exceed a user-specified bound $\alpha$, *i.e.*, $\mathbb{E}[\ell(\mathcal{C}_\lambda(X_{n+1}), Y_{n+1})] \leq \alpha$. However, in our framework, we re-define the loss $\ell$ so that it does not only depend on the prediction set and the label value but also on the expert's prediction on their own.

Under the counterfactual monotonicity assumption, we set the value of the loss $\ell$ using the expression of counterfactual harm in Eq. 8 and, using a similar proof technique as in Angelopoulous et al., first prove the following theorem:

**Theorem 1** *Let $\mathcal{D} = \{(X_i, Y_i, \hat{Y}_i)\}_{i=1}^n$ be a calibration set, with $(X_i, Y_i, \hat{Y}_i) \sim P^{\mathcal{M}}(X, Y, \hat{Y})$, $\alpha \in [0, 1]$ be a user-specified bound, and*

$$\hat{\lambda}(\alpha) = \inf\left\{\lambda : \frac{n}{n+1}\hat{H}_n(\lambda) + \frac{1}{n+1} \leq \alpha\right\} \text{ where } \hat{H}_n(\lambda) = \frac{\sum_{i=1}^n \mathbb{1}\{\hat{Y}_i = Y_i \wedge Y_i \notin \mathcal{C}_\lambda(X_i)\}}{n}. \tag{10}$$

*If counterfactual monotonicity holds, a test sample $(X_{n+1}, Y_{n+1}, \hat{Y}_{n+1}) \sim P^{\mathcal{M}}(X, Y, \hat{Y})$ satisfies that*

$$\mathbb{E}\left[\mathbb{1}\{\hat{Y}_{n+1} = Y_{n+1} \wedge Y_{n+1} \notin \mathcal{C}_{\hat{\lambda}(\alpha)}(X_{n+1})\}\right] \leq \alpha,$$

*where the expectation is over the randomness in the calibration set used to compute the threshold $\hat{\lambda}(\alpha)$ and the test sample.*

Then, we leverage the above theorem and the fact that, under counterfactual monotonicity, the counterfactual harm is nonincreasing with respect to $\lambda$, as shown in Lemma 3 in Appendix B.5, to recover the largest harm-controlling set $\Lambda(\alpha)$:

**Corollary 1** *Let $\alpha \in [0,1]$ be a user-specified bound. Then, under the counterfactual monotonicity assumption, it holds that $\Lambda(\alpha) = \{\lambda \in [0,1] \,|\, \lambda \geq \hat{\lambda}(\alpha)\}$, where $\hat{\lambda}(\alpha)$ is given by Eq. 10.*

Under the interventional monotonicity assumption, rather than directly controlling the counterfactual harm, we will control the upper bound given by Eq. 9. Consequently, rather than recovering the largest harm-controlling set $\Lambda(\alpha)$, we will recover a harm-controlling set $\Lambda'(\alpha) \subseteq \Lambda(\alpha)$. However, since the expression inside the expectation of the upper bound is nonmonotone with respect to $\lambda$, we cannot directly use it to set the value of the loss $\ell$ in conformal risk control. That said, since the first term of the expression is nonincreasing and the second term is nondecreasing, as shown in Lemmas 3 and 4 in Appendix B.5, we can apply conformal risk control separately for each term. For the first term, we use Theorem 1 because the term matches the counterfactual harm under the counterfactual monotonicity assumption. For the second term, we prove the following theorem:

**Theorem 2** *Let $\mathcal{D} = \{(X_i, Y_i, \hat{Y}_i)\}_{i=1}^n$ be a calibration set, with $(X_i, Y_i, \hat{Y}_i) \sim P^{\mathcal{M}}$, $\alpha \in \left[\frac{1}{n+1}, 1\right]$ be a user-specified bound, and*

$$\check{\lambda}(\alpha) = \sup\left\{\lambda \,:\, \frac{n}{n+1}\hat{G}_n(\lambda) + \frac{1}{n+1} \leq \alpha\right\} \ \text{ where } \hat{G}_n(\lambda) = \frac{\sum_{i=1}^n \mathbb{1}\{\hat{Y}_i \neq Y_i \wedge Y_i \in \mathcal{C}_\lambda(X_i)\}}{n}.$$

(11)

*If interventional monotonicity holds and $\check{\lambda}$ exists, a test sample $(X_{n+1}, Y_{n+1}, \hat{Y}_{n+1}) \sim P^{\mathcal{M}}(X, Y, \hat{Y})$ satisfies that*

$$\mathbb{E}\left[\mathbb{1}\{\hat{Y}_{n+1} \neq Y_{n+1} \wedge Y_{n+1} \in \mathcal{C}_{\check{\lambda}(\alpha)}(X_{n+1})\}\right] \leq \alpha,$$

*where the expectation is over the randomness in the calibration set used to compute the threshold $\check{\lambda}(\alpha)$ and the test sample $(X_{n+1}, Y_{n+1}, \hat{Y}_{n+1})$.*

Finally, we leverage the above theorems and the fact that the first term inside the expectation of the upper bound of counterfactual harm in Eq. 9 is nonincreasing and the second term is nondecreasing with respect to $\lambda$ to recover a harm-controlling set $\Lambda'(\alpha) \subseteq \Lambda(\alpha)$:

**Corollary 2** *Let $\alpha \in [0,1]$ be a user-specified bound. Then, under the interventional monotonicity assumption, for any choice of $\alpha' \leq \alpha$, the set*

$$\Lambda'(\alpha) = \{\lambda \in [0,1] \,|\, \hat{\lambda}(\alpha') \leq \lambda \leq \check{\lambda}(\alpha - \alpha')\},$$

*where $\hat{\lambda}(\alpha')$ is given by Eq. 10 and $\check{\lambda}(\alpha - \alpha')$ is given by Eq. 11 satisfies that $\Lambda'(\alpha) \subseteq \Lambda(\alpha)$.*

Note that, in practice, the above corollaries can be implemented efficiently since, by definition, the functions $\hat{H}_n(\lambda)$ and $\hat{G}_n(\lambda)$ are piecewise constant functions of $\lambda$ with $n$ different pieces.

## 6   Experiments

In this section, we use data from two different human subject studies to: a) evaluate the average counterfactual harm caused by decision support systems based on prediction sets; b) validate the theoretical guarantees offered by our computational framework (*i.e.*, Corollaries 1 and 2); c) investigate the trade-off between the average counterfactual harm caused by decision support systems based on prediction sets and the average accuracy achieved by human experts using these systems. In what follows, we assume that counterfactual monotonicity holds. In Appendix G, we conduct experiments where we relax this assumption and assume instead that interventional monotonicity holds.[7]

**Experimental setup.** We first experiment with the ImageNet16H dataset by Steyvers et al. [52], which comprises 32,431 predictions made by 145 human participants on their own about noisy

---

[7]All experiments ran on a Mac OS machine with an M1 processor and 16GB Memory. An open-source implementation of our methodology is publicly available at https://github.com/Networks-Learning/controlling-counterfactual-harm-prediction-sets.

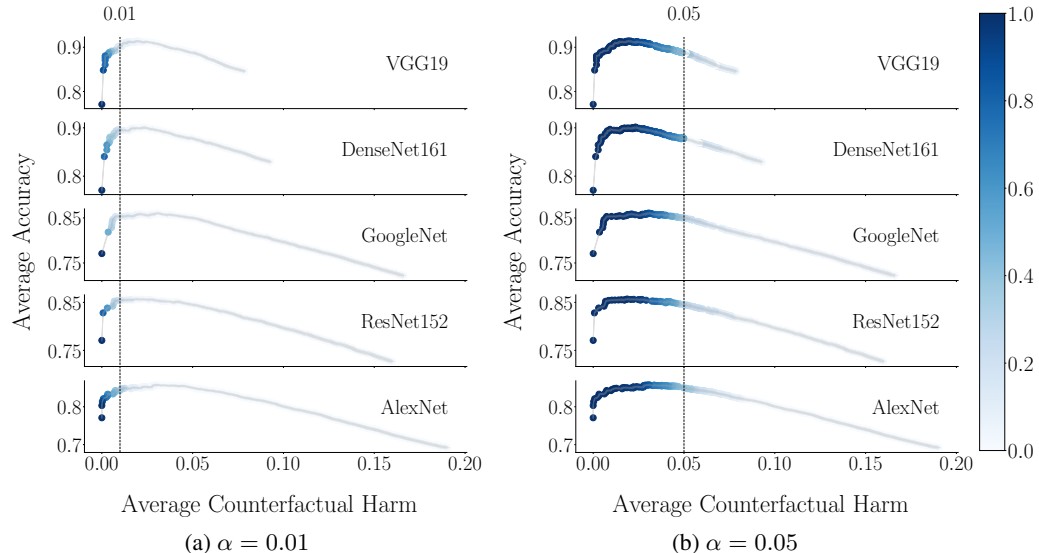

Figure 2: Average accuracy estimated by the mixture of MNLs against the average counterfactual harm for images with $\omega = 110$. Each point corresponds to a $\lambda$ value from 0 to 1 with step 0.001 and the coloring indicates the relative frequency with which each $\lambda$ value is in $\Lambda(\alpha)$ across random samplings of the calibration set. Each row corresponds to decision support systems $\mathcal{C}_\lambda$ with a different pre-trained classifier with average accuracies 0.846 (VGG19), 0.830 (DenseNet), 0.722 (GoogleNet), 0.727 (ResNet152), and 0.691 (AlexNet). The average accuracy achieved by the simulated human experts on their own is 0.771. The results are averaged across 50 random samplings of the test and calibration set. In both panels, 95% confidence intervals are represented using shaded areas and always have width below 0.02.

images created using 1,200 unique natural images from the ImageNet Large Scale Visual Recognition Challenge (ILSRVR) 2012 dataset [53]. More specifically, each of the natural images was used to generate four noisy images with different amount of phase noise distortion $\omega \in \{80, 95, 110, 125\}$ and with the same ground-truth label $y$ from a label set $\mathcal{Y}$ of size $L = 16$. Here, the amount of phase noise controls the difficulty of the classification task—the higher the noise, the more difficult the classification task. In our experiments, we use (all) the noisy images with noise value $\omega \in \{80, 95, 110\}$ because, for the noisy images with $\omega = 125$, humans perform poorly; moreover, we stratify these images (and human predictions) with respect to their amount of phase noise.

For each stratum of images, we apply our framework to decision support systems $\mathcal{C}_\lambda$ with different pre-trained classifiers, namely VGG19 [54], DenseNet161 [55], GoogleNet [56], ResNet152 [57] and AlexNet [58] after 10 epochs of fine-tuning, as provided by Steyvers et al. [52].[8] To this end, we randomly split the images (and human predictions) into a calibration set (10%), which we use to find the harm-controlling sets $\Lambda(\alpha)$ by applying Corollary 1, and a test set (90%), which we use to estimate the average counterfactual harm $H(\lambda)$ caused by the decision support systems $\mathcal{C}_\lambda$ as well as the average accuracy $A(\lambda)$ of the predictions made by a human expert using $\mathcal{C}_\lambda$. Here, since the dataset only contains predictions made by human participants on their own, we use the mixture of multinomial logit models (MNLs) introduced by Straitouri et al. [17] to estimate the average accuracy $A(\lambda)$. Refer to Appendix C for more details regarding the mixture of MNLs and to Appendix D for additional experiments studying the relationship between average counterfactual harm $H(\lambda)$, average prediction set size and empirical coverage[9].

Then, we experiment with the ImageNet16H-PS dataset[10] by Straitouri et al. [18], which comprises 194,407 predictions made by 2,751 human participants using decision support systems $\mathcal{C}_\lambda$ about the

---

[8]All classifiers and images are publicly available at https://osf.io/2ntrf.

[9]By empirical coverage, we refer to the fraction of the test samples for which the prediction sets include the ground truth label value.

[10]The dataset is publicly available at https://github.com/Networks-Learning/counterfactual-prediction-sets/.

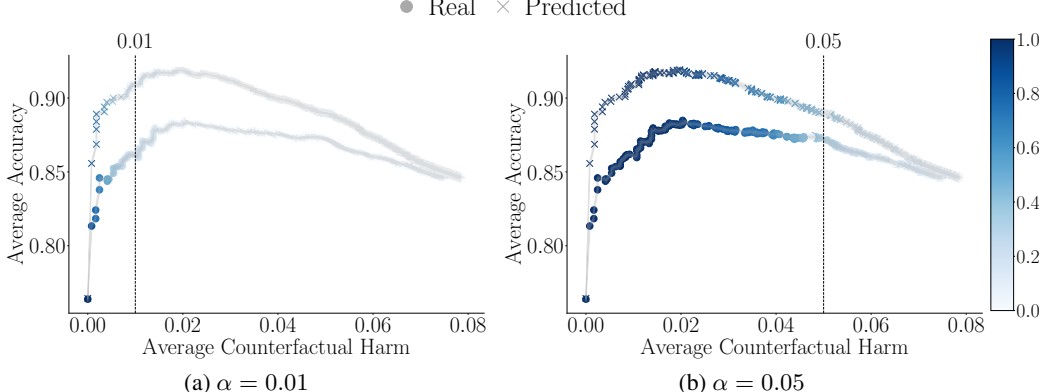

(a) $\alpha = 0.01$  (b) $\alpha = 0.05$

Figure 3: Average accuracy estimated using predictions by human participants (Real) and using the mixture of MNLs (Predicted) against the average counterfactual harm for images with $\omega = 110$. Each point corresponds to a $\lambda$ value from 0 to 1 with step 0.001 and the coloring indicates the relative frequency with which the $\lambda$ value is in $\Lambda(\alpha)$ across random samplings of the calibration set. The decision support systems $\mathcal{C}_\lambda$ use the pre-trained classifier VGG19. The results are averaged across 50 random samplings of the test and calibration set. In both panels, 95% confidence intervals are represented using shaded areas and always have width below 0.02.

set of noisy images with $\omega = 110$ described above. More specifically, for each noisy image, the dataset contains human predictions made under any possible prediction set that can be constructed using Eq. 1 with (the softmax output of) VGG19 after 10 epochs of fine-tuning, as provided by Steyvers et al. [52]. Here, similarly as in the ImageNet16H dataset, we randomly split the images (and human predictions) into a calibration set (10%), which we use to find $\Lambda(\alpha)$, and a test set (90%), which we use to estimate $H(\lambda)$ and $A(\lambda)$. However, in this case, we can estimate $A(\lambda)$ using the predictions made by human participants using $\mathcal{C}_\lambda$ from the dataset, and we can compare this empirical estimate to the one using the mixture of MNLs.

In both datasets, we calculate confidence intervals and validate the theoretical guarantees offered by Corollary 1 by repeating each experiment 50 times and, each time, sampling different calibration and test sets.

**Results.** Figure 2 shows the average accuracy $A(\lambda)$ achieved by a human participant using $\mathcal{C}_\lambda$, as predicted by the mixture of MNLs, against the average counterfactual harm $H(\lambda)$ caused by $\mathcal{C}_\lambda$ for $\alpha \in \{0.01, 0.05\}$ on the stratum of images with $\omega = 110$ from the ImageNet16H dataset. Refer to Appendix D for results on other strata. Here, each point corresponds to a different $\lambda$ value and its coloring indicates the empirical probability that $\lambda$ is included in the harm-controlling set $\Lambda(\alpha)$. The results show several interesting insights. First, we find that, as long as $\lambda < 1$, the decision support systems $\mathcal{C}_\lambda$ always cause some amount of counterfactual harm[11]. This suggests that, while restricting human agency enables human-AI complementarity, it inevitably causes (some amount of) counterfactual harm. Second, we find that the sets of $\lambda$ values provided by our framework are typically harm-controlling, *i.e.*, they do not include $\lambda$ values such that $H(\lambda) > \alpha$. However, the sets are often conservative and do not include *all* the harm-controlling $\lambda$ values due to estimation error in the the empirical estimate $\hat{H}_n(\lambda)$ of the average counterfactual harm using data from the calibration set. In Appendix D, we show that using larger calibration sets reduces the above mentioned estimation error and results in sets of $\lambda$ values that are less conservative. Third, we find that there is a trade-off between accuracy and counterfactual harm and this trade-off is qualitatively consistent across decision support systems using different pre-trained classifiers. In fact, for $\alpha = 0.01$, the decision support systems $\mathcal{C}_{\lambda^*}$ offering the greatest average accuracy $A(\lambda^*)$ are not harm-controlling.

Figure 3 shows the average accuracy $A(\lambda)$ achieved by a human participant using $\mathcal{C}_\lambda$, as estimated using predictions made by human participants using $\mathcal{C}_\lambda$, against the average counterfactual harm

---

[11]For $\lambda = 1$, the decision support system $\mathcal{C}_\lambda$ does not cause harm because the prediction sets always contain all label values and thus human participants always make predictions on their own.

$H(\lambda)$ caused by $\mathcal{C}_\lambda$ for $\alpha \in \{0.01, 0.05\}$ on the ImageNet16H-PS dataset. Here, the meaning and coloring of each point is the same as in Figure 2. The results also support the findings derived from the experiments with the ImageNet16H dataset—the decision support systems $\mathcal{C}_\lambda$ always cause some amount of harm, our framework succeeds at identifying harm-controlling sets, and there is a trade-off between accuracy and counterfactual harm. Further, they also show that, while there is a gap between the average accuracy estimated using the mixture of MNLs and the average accuracy estimated using predictions made by human participants using $\mathcal{C}_\lambda$, they follow the same trend and support the same qualitative conclusions.

## 7 Discussion and limitations

In this section, we highlight several limitations of our work, discuss its broader impact, and propose avenues for future work.

**Assumptions.** We have assumed that the data samples and the expert predictions are drawn i.i.d. from a fixed distribution and the calibration set contains samples with noiseless ground truth labels and expert label predictions. It would be very interesting to extend our framework to allow for distribution shifts and label noise. Moreover, we have considered prediction sets constructed with a *fixed* user-specified threshold value $\lambda$. In light of recent work by Gibbs et al. [59], it would be interesting to extend our framework to allow for threshold values that depend on the data samples. Furthermore, in our definition of counterfactual harm we have treated all inaccurate predictions as equally harmful. Expanding our definition of harm to weigh different inaccurate predictions according to their consequences would be a very interesting avenue for future work. In addition, under the interventional monotonicity assumption, we have empirically observed that the gap between our lower- and upper-bounds is often large. Therefore, it would be useful to identify other natural assumptions under which this gap is smaller.

**Methodology.** In our framework, the prediction sets are constructed using the softmax output of a pre-trained classifier. However, we hypothesize that, by accounting for the similarity between the mistakes made by humans and those made by the pre-trained classifier, we may be able to construct prediction sets that cause counterfactual harm less frequently. Moreover, we have focused on controlling the average counterfactual harm. However, whenever the expert's predictions are consequential to individuals, this may lead significant disparities across demographic groups. Therefore, it would be important to extend our framework to account for fairness considerations.

**Evaluation.** Our experimental evaluation comprises a single benchmark dataset of noisy natural images and thus one may question the generalizability of the conclusions we draw from our results. To overcome this limitation, it would be important to further investigate the trade-off between accuracy and counterfactual harm in decision support systems based on prediction sets in real-world application domains (*e.g.*, medical diagnosis).

## 8 Conclusions

In this paper, we have initiated the study of counterfactual harm in decision support systems based on prediction sets. We have introduced a computational framework that, under natural monotonicity assumptions on the predictions made by experts using these systems, can control how frequently these systems cause harm. Moreover, we have validated our framework using data from two different human subject studies and shown that, in decision support systems based on prediction sets, there is a trade-off between accuracy and counterfactual harm.

## Acknowledgments and Disclosure of Funding

Gomez-Rodriguez acknowledges support from the European Research Council (ERC) under the European Union's Horizon 2020 research and innovation programme (grant agreement No. 945719).

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

# A  Comparison to Pearl's notation

Using Pearl's notation [50], our definition of counterfactual harm reads as follows:

$$h_\lambda(x, y, \hat{y}) = \mathbb{E}[\max\{0, \mathbb{1}\{\hat{y} = y\} - \mathbb{1}\{\hat{Y}_\lambda = y\}\}|X = x, Y = y, \hat{Y}_1 = \hat{y}],$$

where the subindices in $\hat{Y}_\lambda$ and $\hat{Y}_1$ denote (hard) interventions on the exogenous variable $\Lambda$, the random variables $x, y, \hat{y} \sim P(X, Y, \hat{Y}_1)$, and the expectation is over the prediction $\hat{Y}_\lambda$ made by the expert using the system $\mathcal{C}_\lambda$. Using the above notation, the definition does not explicitly specify the (counterfactual) distribution used in the expectation, in contrast to the definition using the notation by Peters et al. [51], which explicitly specifies the counterfactual distribution used in the expectation, as restated below for clarity.

$$h_\lambda(x, y, \hat{y}) = \mathbb{E}_{\hat{Y} \sim P^{\mathcal{M};\mathrm{do}(\Lambda=\lambda) \mid \hat{Y}=\hat{y}, X=x, Y=y}(\hat{Y})}[\max\{0, \mathbb{1}\{\hat{y} = y\} - \mathbb{1}\{\hat{Y} = y\}\}].$$

# B Proofs

## B.1 Proof of Proposition 1

Given an observation $(x, y, \hat{y})$ under no intervention, we have

$$
\begin{aligned}
h_\lambda(x, y, \hat{y}) &= \mathbb{E}_{\hat{Y} \sim P^{\mathcal{M};\mathrm{do}(\Lambda=\lambda) \mid \hat{Y}=\hat{y}, X=x, Y=y}}[\max\{0, \mathbb{1}\{\hat{y} = y\} - \mathbb{1}\{\hat{Y} = y\}\}] \\
&= \mathbb{E}_{\hat{Y} \sim P^{\mathcal{M};\mathrm{do}(\Lambda=\lambda) \mid \hat{Y}=\hat{y}, X=x, Y=y}}[\max\{0, 1 - \mathbb{1}\{\hat{Y} = y\}\}] \cdot \mathbb{1}\{\hat{y} = y\} \\
&\quad + \mathbb{E}_{\hat{Y} \sim P^{\mathcal{M};\mathrm{do}(\Lambda=\lambda) \mid \hat{Y}=\hat{y}, X=x, Y=y}}[\max\{0, 0 - \mathbb{1}\{\hat{Y} = y\}\}] \cdot \mathbb{1}\{\hat{y} \neq y\} \\
&= \mathbb{E}_{\hat{Y} \sim P^{\mathcal{M};\mathrm{do}(\Lambda=\lambda) \mid \hat{Y}=\hat{y}, X=x, Y=y}}[\max\{0, 1 - \mathbb{1}\{\hat{Y} = y\}\}] \cdot \mathbb{1}\{\hat{y} = y\} + 0 \\
&= \mathbb{E}_{\hat{Y} \sim P^{\mathcal{M};\mathrm{do}(\Lambda=\lambda) \mid \hat{Y}=\hat{y}, X=x, Y=y}}[\max\{0, 1 - 1\}] \cdot \mathbb{1}\{\hat{y} = y\} \cdot \mathbb{1}\{y \in \mathcal{C}_\lambda(x)\} \\
&\quad + \mathbb{E}_{\hat{Y} \sim P^{\mathcal{M};\mathrm{do}(\Lambda=\lambda) \mid \hat{Y}=\hat{y}, X=x, Y=y}}[\max\{0, 1 - 0\}] \cdot \mathbb{1}\{\hat{y} = y\} \cdot \mathbb{1}\{y \notin \mathcal{C}_\lambda(x)\} \quad (12) \\
&= 0 \cdot \mathbb{1}\{\hat{y} = y\} \cdot \mathbb{1}\{y \in \mathcal{C}_\lambda(x)\} \\
&\quad + \mathbb{E}_{\hat{Y} \sim P^{\mathcal{M};\mathrm{do}(\Lambda=\lambda) \mid \hat{Y}=\hat{y}, X=x, Y=y}}[\max\{0, 1 - 0\}] \cdot \mathbb{1}\{\hat{y} = y\} \cdot \mathbb{1}\{y \notin \mathcal{C}_\lambda(x)\} \\
&= 1 \cdot \mathbb{1}\{\hat{y} = y\} \cdot \mathbb{1}\{y \notin \mathcal{C}_\lambda(x)\} \\
&= \mathbb{1}\{\hat{y} = y\} \cdot \mathbb{1}\{y \notin \mathcal{C}_\lambda(x)\} \\
&= \mathbb{1}\{\hat{y} = y \wedge y \notin \mathcal{C}_\lambda(x)\},
\end{aligned}
$$

where, in Eq. 12, we used the counterfactual monotonicity property by setting $\mathbb{1}\{\hat{Y} = y\} = 1$ if $y \in \mathcal{C}_\lambda(x)$.

## B.2 Proof of Proposition 2

Under interventional monotonicity, by Lemma 1, we have

$$
\begin{aligned}
h_\lambda(x, y, \hat{y}) &= \mathbb{E}_{\hat{Y} \sim P^{\mathcal{M};\mathrm{do}(\Lambda=\lambda) \mid \hat{Y}=\hat{y}, X=x, Y=y}}[\mathbb{1}\{\hat{Y} \neq y\}] \cdot \mathbb{1}\{\hat{y} = y\} \cdot \mathbb{1}\{y \in \mathcal{C}_\lambda(x)\} \\
&\quad + \mathbb{1}\{\hat{y} = y\} \cdot \mathbb{1}\{y \notin \mathcal{C}_\lambda(x)\} \\
&\leq \mathbb{E}_{\hat{Y} \sim P^{\mathcal{M};\mathrm{do}(\Lambda=\lambda) \mid \hat{Y}=\hat{y}, X=x, Y=y}}[\mathbb{1}\{\hat{Y} \neq y\}] \cdot \mathbb{1}\{y \in \mathcal{C}_\lambda(x)\} \\
&\quad + \mathbb{1}\{\hat{y} = y\} \cdot \mathbb{1}\{y \notin \mathcal{C}_\lambda(x)\},
\end{aligned}
$$

where we used that $\mathbb{1}\{\hat{y} = y\} \leq 1$.

By taking the expectation on both sides over $\hat{y}$ under no intervention we have

$$
\mathbb{E}_{\hat{Y}' \sim P^{\mathcal{M}}} \left[ h_\lambda(x, y, \hat{Y}') \right] \leq \mathbb{E}_{\hat{Y}' \sim P^{\mathcal{M}}} \left[ \mathbb{E}_{\hat{Y} \sim P^{\mathcal{M};\mathrm{do}(\Lambda=\lambda) \mid \hat{Y}=\hat{Y}', X=x, Y=y}} \left[ \mathbb{1}\{\hat{Y} \neq y\} \right] \cdot \mathbb{1}\{y \in \mathcal{C}_\lambda(x)\} \right]
$$

$$
(13)
$$

$$
+ \mathbb{E}_{\hat{Y}' \sim P^{\mathcal{M}}} \left[ \mathbb{1}\{\hat{Y}' = y\} \cdot \mathbb{1}\{y \notin \mathcal{C}_\lambda(x)\} \right] \tag{14}
$$

$$
= \mathbb{E}_{\hat{Y}' \sim P^{\mathcal{M}}} \left[ \mathbb{E}_{\hat{Y} \sim P^{\mathcal{M};\mathrm{do}(\Lambda=\lambda) \mid \hat{Y}=\hat{Y}', X=x, Y=y}} \left[ \mathbb{1}\{\hat{Y} \neq y\} \right] \right] \cdot \mathbb{1}\{y \in \mathcal{C}_\lambda(x)\} \tag{15}
$$

$$
+ \mathbb{E}_{\hat{Y}' \sim P^{\mathcal{M}}} \left[ \mathbb{1}\{\hat{Y}' = y\} \cdot \mathbb{1}\{y \notin \mathcal{C}_\lambda(x)\} \right], \tag{16}
$$

since $\mathbb{1}\{y \in \mathcal{C}_\lambda(x)\}$ is a constant with respect to the expectation over $\hat{Y}' \sim P^{\mathcal{M}}$.

By Lemma 2, Eq. 13 becomes

$$
\mathbb{E}_{\hat{Y}' \sim P^{\mathcal{M}}} \left[ h_\lambda(x, y, \hat{Y}') \right] \leq \mathbb{E}_{\hat{Y} \sim P^{\mathcal{M};\mathrm{do}(\Lambda=\lambda)}} \left[ \mathbb{1}\{\hat{Y} \neq Y\} \mid X = x, Y = y \right] \cdot \mathbb{1}\{y \in \mathcal{C}_\lambda(x)\}
$$

$$
+ \mathbb{E}_{\hat{Y}' \sim P^{\mathcal{M}}} \left[ \mathbb{1}\{\hat{Y}' = y\} \cdot \mathbb{1}\{y \notin \mathcal{C}_\lambda(x)\} \right]. \tag{17}
$$

By Assumption 2 for $\lambda' = 1$ and any $\lambda$ such that $y \in \mathcal{C}_\lambda(x)$, we have that

$$
P^{\mathcal{M};\mathrm{do}(\Lambda=\lambda)}(\hat{Y} = Y \mid X = x, Y = y) \geq P^{\mathcal{M}}(\hat{Y} = Y \mid X = x, Y = y),
$$

which we can rewrite as

$$
\mathbb{E}_{\hat{Y} \sim P^{\mathcal{M};\mathrm{do}(\Lambda=\lambda)}}[\mathbb{1}\{\hat{Y} = Y\} \mid X = x, Y = y] \geq \mathbb{E}_{\hat{Y} \sim P^{\mathcal{M}}}[\mathbb{1}\{\hat{Y} = Y\} \mid X = x, Y = y],
$$

and is equivalent to

$$\mathbb{E}_{\hat{Y} \sim P^{\mathcal{M};\mathrm{do}(\Lambda=\lambda)}}[\mathbb{1}\{\hat{Y} \neq Y\} \mid X = x, Y = y] \leq \mathbb{E}_{\hat{Y} \sim P^{\mathcal{M}}}[\mathbb{1}\{\hat{Y} \neq Y\} \mid X = x, Y = y].$$

Using the above result, Eq. 17 becomes

$$\mathbb{E}_{\hat{Y}' \sim P^{\mathcal{M}}}\left[h_\lambda(x, y, \hat{Y}')\right] \leq \mathbb{E}_{\hat{Y} \sim P^{\mathcal{M}}}\left[\mathbb{1}\{\hat{Y} \neq Y\} \mid X = x, Y = y\right] \cdot \mathbb{1}\{y \in \mathcal{C}_\lambda(x)\}$$
$$+ \mathbb{E}_{\hat{Y}' \sim P^{\mathcal{M}}}\left[\mathbb{1}\{\hat{Y}' = y\} \cdot \mathbb{1}\{y \notin \mathcal{C}_\lambda(x)\}\right].$$

If we now take the expectation in both sides over $X, Y \sim P^{\mathcal{M}}$ and use that $\hat{Y}'$ is equal in distribution with $\hat{Y}$, we have

$$\mathbb{E}_{X,Y,\hat{Y} \sim P^{\mathcal{M}}}\left[h_\lambda(X, Y, \hat{Y})\right] \leq \mathbb{E}_{X,Y,\hat{Y} \sim P^{\mathcal{M}}}\left[\mathbb{1}\{\hat{Y} \neq Y\} \cdot \mathbb{1}\{Y \in \mathcal{C}_\lambda(X)\}\right]$$
$$+ \mathbb{E}_{X,Y,\hat{Y} \sim P^{\mathcal{M}}}\left[\mathbb{1}\{\hat{Y} = Y\} \cdot \mathbb{1}\{Y \notin \mathcal{C}_\lambda(X)\}\right],$$

that is equivalent to

$$\mathbb{E}_{X,Y,\hat{Y} \sim P^{\mathcal{M}}}\left[h_\lambda(X, Y, \hat{Y})\right]$$
$$\leq \mathbb{E}_{X,Y,\hat{Y} \sim P^{\mathcal{M}}}\left[\mathbb{1}\{\hat{Y} \neq Y\} \cdot \mathbb{1}\{Y \in \mathcal{C}_\lambda(X)\} + \mathbb{1}\{\hat{Y} = Y\} \cdot \mathbb{1}\{Y \notin \mathcal{C}_\lambda(X)\}\right],$$

which we can write as

$$\mathbb{E}_{X,Y,\hat{Y} \sim P^{\mathcal{M}}}\left[h_\lambda(X, Y, \hat{Y})\right]$$
$$\leq \mathbb{E}_{X,Y,\hat{Y} \sim P^{\mathcal{M}}}\left[\mathbb{1}\{\hat{Y} \neq Y \wedge Y \in \mathcal{C}_\lambda(X)\} + \mathbb{1}\{\hat{Y} = Y \wedge Y \notin \mathcal{C}_\lambda(X)\}\right].$$

For the lower-bound, by Lemma 1, we have

$$h_\lambda(x, y, \hat{y}) = \mathbb{E}_{\hat{Y} \sim P^{\mathcal{M};\mathrm{do}(\Lambda=\lambda)} \mid \hat{Y}=\hat{y}, X=x, Y=y}[\mathbb{1}\{\hat{Y} \neq y\}] \cdot \mathbb{1}\{\hat{y} = y\} \cdot \mathbb{1}\{y \in \mathcal{C}_\lambda(x)\}$$
$$+ \mathbb{1}\{\hat{y} = y\} \cdot \mathbb{1}\{y \notin \mathcal{C}_\lambda(x)\}$$
$$\geq \mathbb{1}\{\hat{y} = y\} \cdot \mathbb{1}\{y \notin \mathcal{C}_\lambda(x)\},$$

where we used that $\mathbb{E}_{\hat{Y} \sim P^{\mathcal{M};\mathrm{do}(\Lambda=\lambda)} \mid \hat{Y}=\hat{y}, X=x, Y=y}[\mathbb{1}\{\hat{Y} \neq y\}] \cdot \mathbb{1}\{\hat{y} = y\} \cdot \mathbb{1}\{y \in \mathcal{C}_\lambda(x)\} \geq 0$. By taking the expectation in both sides over $X, Y, \hat{Y} \sim P^{\mathcal{M}}$, we have

$$\mathbb{E}_{X,Y,\hat{Y} \sim P^{\mathcal{M}}}\left[\mathbb{1}\{\hat{Y} = Y\} \cdot \mathbb{1}\{Y \notin \mathcal{C}_\lambda(X)\}\right] \leq \mathbb{E}_{X,Y,\hat{Y} \sim P^{\mathcal{M}}}\left[h_\lambda(X, Y, \hat{Y})\right],$$

that is equivalent to

$$\mathbb{E}_{X,Y,\hat{Y} \sim P^{\mathcal{M}}}\left[\mathbb{1}\{\hat{Y} = Y \wedge Y \notin \mathcal{C}_\lambda(X)\}\right] \leq \mathbb{E}_{X,Y,\hat{Y} \sim P^{\mathcal{M}}}\left[h_\lambda(X, Y, \hat{Y})\right],$$

thus we conclude the proof.

### B.3 Proof of Theorem 1

We will show that $\mathbb{1}\{\hat{Y}_i = Y_i \wedge Y_i \notin \mathcal{C}_\lambda(X_i)\}$ are exchangeable and that they satisfy the requirements of Theorem 1 in [22] for each $X_i, Y_i, \hat{Y}_i$. Since $X_i, Y_i, \hat{Y}_i$ are i.i.d., $\mathbb{1}\{\hat{Y}_i = Y_i \wedge Y_i \notin \mathcal{C}_\lambda(X_i)\}$ are also i.i.d thus exchangeable. Considering each $\mathbb{1}\{\hat{Y}_i = Y_i \wedge Y_i \notin \mathcal{C}_\lambda(X_i)\}$ for each $X_i, Y_i, \hat{Y}_i \in \mathcal{D}$ as a function of $\lambda$, it holds by Lemma 3 that for each $X_i, Y_i, \hat{Y}_i$, $\mathbb{1}\{\hat{Y}_i = Y_i \wedge Y_i \notin \mathcal{C}_\lambda(X_i)\}$ is non-increasing in $\lambda$ and right-continuous. In addition, it holds that $\mathbb{1}\{\hat{Y}_i = Y_i \wedge Y_i \notin \mathcal{C}_1(X_i)\} = 0 \leq \alpha$ and that $\sup_\lambda \mathbb{1}\{\hat{Y}_i = Y_i \wedge Y_i \notin \mathcal{C}_\lambda(X_i)\} \leq 1$ surely by definition.

## B.4 Proof of Theorem 2

For the proof we will follow a similar procedure as in [22]. Since $\mathbb{1}\{\hat{Y} \neq Y \wedge Y \in \mathcal{C}_\lambda(X)\} \leq 1$, $\forall X, Y, \hat{Y}, \lambda$, it will hold that

$$\hat{G}_{n+1}(\lambda) = \frac{n\hat{G}_n(\lambda) + \mathbb{1}\{\hat{Y}_{n+1} \neq Y_{n+1} \wedge Y_{n+1} \in \mathcal{C}_\lambda(X_{n+1})\}}{n+1} \leq \frac{n}{n+1}\hat{G}_n(\lambda) + \frac{1}{n+1}, \quad (18)$$

for every $\lambda \in \mathcal{L}$. Now, let

$$\hat{\lambda}' = \sup\{\lambda \in \mathcal{L} : \hat{G}_{n+1}(\lambda) \leq \alpha\}. \quad (19)$$

If such $\hat{\lambda}'$ does not exist, by Eq. 19 $\hat{\lambda}$ will also not exist. If $\hat{\lambda}'$ exists,

we will show that $\hat{\lambda} \leq \hat{\lambda}'$. If $\hat{\lambda} = 0$, then it will hold that $\hat{\lambda}' \geq \hat{\lambda} = 0 = \min_{\lambda \in \mathcal{L}} \lambda$. If $\hat{\lambda} > 0$, then by Eq. 19, $\hat{\lambda}$ must satisfy

$$\hat{G}_{n+1}(\hat{\lambda}) \leq \frac{n}{n+1}\hat{G}_n(\hat{\lambda}) + \frac{1}{n+1} \leq \alpha, \quad (20)$$

which means that $\hat{\lambda} \leq \hat{\lambda}'$. Therefore, in any case we have that $\hat{\lambda} \leq \hat{\lambda}'$.

From Lemma 4, the random functions $\mathbb{1}\{\hat{Y}_i \neq Y_i \wedge Y_i \in \mathcal{C}_\lambda(X_i)\}$ are non-decreasing for each $i \in [n+1]$. As a result, since $\hat{\lambda} \leq \hat{\lambda}'$, we have that

$$\mathbb{E}_{X_i, Y_i, \hat{Y}_i \sim P^{\mathcal{M}}}[\mathbb{1}\{\hat{Y}_{n+1} \neq Y_{n+1} \wedge Y_{n+1} \in \mathcal{C}_{\hat{\lambda}}(X_{n+1})\}]$$
$$\leq \mathbb{E}_{X_i, Y_i, \hat{Y}_i \sim P^{\mathcal{M}}}[\mathbb{1}\{\hat{Y}_{n+1} \neq Y_{n+1} \wedge Y_{n+1} \in \mathcal{C}_{\hat{\lambda}'}(X_{n+1})\}], \quad (21)$$

where $i = \{1, \ldots, n+1\}$. Given that the data samples $(X_i, Y_i, \hat{Y}_i)$ for $i \in [1, n+1]$ are exchangeable, the values $\mathbb{1}\{\hat{Y}_i \neq Y_i \wedge Y_i \in \mathcal{C}_{\hat{\lambda}'}(X_i)\}$ for $i \in [n+1]$ are also exchangeable. Therefore, due to exchangeability and given that $\hat{\lambda}'$ is fixed given $(X_i, Y_i, \hat{Y}_i)$ for $i \in [1, n+1]$, $\mathbb{1}\{\hat{Y}_{n+1} \neq Y_{n+1} \wedge Y_{n+1} \in \mathcal{C}_{\hat{\lambda}'}(X_{n+1})\}$ has the same probability of taking any value in $\mathcal{D}_{\hat{\lambda}'} = \{\mathbb{1}\{\hat{Y}_i \neq Y_i \wedge Y_i \in \mathcal{C}_{\hat{\lambda}'}(X_i)\}\}_{i=1}^{n+1}$, *i.e.*, the value of $\mathbb{1}\{\hat{Y}_i \neq Y_i \wedge Y_i \in \mathcal{C}_{\hat{\lambda}'}(X_i)\}$ for the $n+1$-th sample in $\{(X_i, Y_i, \hat{Y}_i)\}_{i=1}^{n+1}$ follows a uniform distribution over all the possible values in $\mathcal{D}_{\hat{\lambda}'}$. As a result, from the law of total expectation it holds for $i = \{1, \ldots, n+1\}$ that $\mathbb{E}_{X_i, Y_i, \hat{Y}_i \sim P^{\mathcal{M}}}[\mathbb{1}\{\hat{Y}_{n+1} \neq Y_{n+1} \wedge Y_{n+1} \in \mathcal{C}_{\hat{\lambda}'}(X_{n+1})\}] = \mathbb{E}_{\mathcal{D}_{\hat{\lambda}'}}[\mathbb{1}\{\hat{Y}_{n+1} \neq Y_{n+1} \wedge Y_{n+1} \in \mathcal{C}_{\hat{\lambda}'}(X_{n+1})\} \mid \mathcal{D}_{\hat{\lambda}'}] = \frac{\sum_{i=1}^{n+1} \mathbb{1}\{\hat{Y}_i \neq Y_i \wedge Y_i \in \mathcal{C}_{\hat{\lambda}'}(X_i)\}}{n+1} = \hat{G}_{n+1}(\hat{\lambda}')$. Finally, from Eq. 20 and the above we have

$$\mathbb{E}_{X_i, Y_i, \hat{Y}_i \sim P^{\mathcal{M}}}[\mathbb{1}\{\hat{Y}_{n+1} \neq Y_{n+1} \wedge Y_{n+1} \in \mathcal{C}_{\hat{\lambda}}(X_{n+1})\}]$$
$$\leq \mathbb{E}_{X_i, Y_i, \hat{Y}_i \sim P^{\mathcal{M}}}[\mathbb{1}\{\hat{Y}_{n+1} \neq Y_{n+1} \wedge Y_{n+1} \in \mathcal{C}_{\hat{\lambda}'}(X_{n+1})\}] = \hat{G}_{n+1}(\hat{\lambda}') \leq \alpha, \quad (22)$$

where $i = \{1, \ldots, n+1\}$.

## B.5 Additional Lemmas

**Lemma 1** *Under the interventional monotonicity assumption, for any $x, y, \hat{y} \sim P^{\mathcal{M}}$, the counterfactual harm that a decision support system $\mathcal{C}_\lambda$ would have caused, if deployed, is given by*

$$h_\lambda(x, y, \hat{y}) = \mathbb{E}_{\hat{Y} \sim P^{\mathcal{M};do(\Lambda=\lambda)} \mid \hat{Y}=\hat{y}, X=x, Y=y}[\mathbb{1}\{\hat{Y} \neq y\}] \cdot \mathbb{1}\{\hat{y} = y\} \cdot \mathbb{1}\{y \in \mathcal{C}_\lambda(x)\}$$
$$+ \mathbb{1}\{\hat{y} = y\} \cdot \mathbb{1}\{y \notin \mathcal{C}_\lambda(x)\}. \quad (23)$$

*Proof.* Given an observation $(x, y, \hat{y})$ under no intervention, we have

$$
\begin{aligned}
h_\lambda(x, y, \hat{y}) &= \mathbb{E}_{\hat{Y} \sim P^{\mathcal{M};\text{do}(\Lambda=\lambda)} \mid \hat{Y}=\hat{y}, X=x, Y=y}[\max\{0, \mathbb{1}\{\hat{y} = y\} - \mathbb{1}\{\hat{Y} = y\}\}] \\
&= \mathbb{E}_{\hat{Y} \sim P^{\mathcal{M};\text{do}(\Lambda=\lambda)} \mid \hat{Y}=\hat{y}, X=x, Y=y}[\max\{0, 1 - \mathbb{1}\{\hat{Y} = y\}\}] \cdot \mathbb{1}\{\hat{y} = y\} \\
&\quad + \mathbb{E}_{\hat{Y} \sim P^{\mathcal{M};\text{do}(\Lambda=\lambda)} \mid \hat{Y}=\hat{y}, X=x, Y=y}[\max\{0, 0 - \mathbb{1}\{\hat{Y} = y\}\}] \cdot \mathbb{1}\{\hat{y} \neq y\} \\
&= \mathbb{E}_{\hat{Y} \sim P^{\mathcal{M};\text{do}(\Lambda=\lambda)} \mid \hat{Y}=\hat{y}, X=x, Y=y}[\max\{0, 1 - \mathbb{1}\{\hat{Y} = y\}\}] \cdot \mathbb{1}\{\hat{y} = y\} + 0 \\
&= \mathbb{E}_{\hat{Y} \sim P^{\mathcal{M};\text{do}(\Lambda=\lambda)} \mid \hat{Y}=\hat{y}, X=x, Y=y}[\max\{0, 1 - \mathbb{1}\{\hat{Y} = y\}\}] \cdot \mathbb{1}\{\hat{y} = y\} \cdot \mathbb{1}\{y \in \mathcal{C}_\lambda(x)\} \\
&\quad + \mathbb{E}_{\hat{Y} \sim P^{\mathcal{M};\text{do}(\Lambda=\lambda)} \mid \hat{Y}=\hat{y}, X=x, Y=y}[\max\{0, 1 - \mathbb{1}\{\hat{Y} = y\}\}] \cdot \mathbb{1}\{\hat{y} = y\} \cdot \mathbb{1}\{y \notin \mathcal{C}_\lambda(x)\} \\
&= \mathbb{E}_{\hat{Y} \sim P^{\mathcal{M};\text{do}(\Lambda=\lambda)} \mid \hat{Y}=\hat{y}, X=x, Y=y}[\max\{0, \mathbb{1}\{\hat{Y} \neq y\}\}] \cdot \mathbb{1}\{\hat{y} = y\} \cdot \mathbb{1}\{y \in \mathcal{C}_\lambda(x)\} \\
&\quad + \mathbb{E}_{\hat{Y} \sim P^{\mathcal{M};\text{do}(\Lambda=\lambda)} \mid \hat{Y}=\hat{y}, X=x, Y=y}[\max\{0, 1 - 0\}] \cdot \mathbb{1}\{\hat{y} = y\} \cdot \mathbb{1}\{y \notin \mathcal{C}_\lambda(x)\} \\
&= \mathbb{E}_{\hat{Y} \sim P^{\mathcal{M};\text{do}(\Lambda=\lambda)} \mid \hat{Y}=\hat{y}, X=x, Y=y}[\max\{0, \mathbb{1}\{\hat{Y} \neq y\}\}] \cdot \mathbb{1}\{\hat{y} = y\} \cdot \mathbb{1}\{y \in \mathcal{C}_\lambda(x)\} \\
&\quad + 1 \cdot \mathbb{1}\{\hat{y} = y\} \cdot \mathbb{1}\{y \notin \mathcal{C}_\lambda(x)\} \\
&= \mathbb{E}_{\hat{Y} \sim P^{\mathcal{M};\text{do}(\Lambda=\lambda)} \mid \hat{Y}=\hat{y}, X=x, Y=y}[\mathbb{1}\{\hat{Y} \neq y\}] \cdot \mathbb{1}\{\hat{y} = y\} \cdot \mathbb{1}\{y \in \mathcal{C}_\lambda(x)\} + \mathbb{1}\{\hat{y} = y\} \cdot \mathbb{1}\{y \notin \mathcal{C}_\lambda(x)\}
\end{aligned}
$$

■

**Lemma 2** *Under the structural causal model $\mathcal{M}$ satisfying Eq. 3, given an observation $(x, y)$ it holds*

$$
\mathbb{E}_{\hat{Y}' \sim P^{\mathcal{M}}} \left[\mathbb{E}_{\hat{Y} \sim P^{\mathcal{M};do(\Lambda=\lambda)} \mid X=x, Y=y, \hat{Y}=\hat{Y}'} \left[\mathbb{1}\{\hat{Y} \neq Y\}\right]\right]
$$
$$
= \mathbb{E}_{\hat{Y} \sim P^{\mathcal{M};do(\Lambda=\lambda)}}[\mathbb{1}\{\hat{Y} \neq Y\} \mid X = x, Y = y]. \quad (24)
$$

*Proof.* Given an observation $(x, y, \hat{y})$ under no intervention, for the expectation of the prediction of the human $\hat{Y} \sim P^{\mathcal{M};\text{do}(\Lambda=\lambda)} \mid X=x, Y=y, \hat{Y}=\hat{y}$ it holds

$$
\mathbb{E}_{\hat{Y} \sim P^{\mathcal{M};\text{do}(\Lambda=\lambda)} \mid X=x, Y=y, \hat{Y}=\hat{y}} \left[\mathbb{1}\{\hat{Y} \neq Y\}\right]
$$
$$
= \sum_{u,v} P(U = u, V = v \mid X = x, Y = y, \hat{Y} = \hat{y}) \cdot \mathbb{1}\{f_{\hat{Y}}(u, v, \mathcal{C}_\lambda(x)) \neq y\},
$$

where $P(U = u, V = v \mid X = x, Y = y, \hat{Y} = \hat{y})$ is the posterior distribution of the exogenous variables $U, V$ conditional on the observation $(x, y, \hat{y})$.

By taking the expectation of the above with respect to the human prediction $\hat{y}$ under no intervention we have

$$
\begin{aligned}
&\mathbb{E}_{\hat{Y}' \sim P^{\mathcal{M}}} \left[\mathbb{E}_{\hat{Y} \sim P^{\mathcal{M};\text{do}(\Lambda=\lambda)} \mid X=x, Y=y, \hat{Y}=\hat{Y}'} \left[\mathbb{1}\{\hat{Y} \neq Y\}\right]\right] \\
&= \sum_{\hat{y}} P^{\mathcal{M}}(\hat{Y}' = \hat{y} \mid X = x, Y = y) \\
&\quad \cdot \sum_{u,v} P(U = u, V = v \mid X = x, Y = y, \hat{Y} = \hat{y}) \cdot \mathbb{1}\{f_{\hat{Y}}(u, v, \mathcal{C}_\lambda(x)) \neq y\} \\
&= \sum_{u,v} \sum_{\hat{y}} P(U = u, V = v \mid X = x, Y = y, \hat{Y} = \hat{y}) \cdot P^{\mathcal{M}}(\hat{Y}' = \hat{y} \mid X = x, Y = y) \\
&\quad \cdot \mathbb{1}\{f_{\hat{Y}}(u, v, \mathcal{C}_\lambda(x)) \neq y\}.
\end{aligned}
$$

Since $\hat{Y}'$ is equal in distribution to $\hat{Y}$ we can rewrite the above as

$$
\begin{aligned}
&\mathbb{E}_{\hat{Y}' \sim P^{\mathcal{M}}} \left[\mathbb{E}_{\hat{Y} \sim P^{\mathcal{M};\text{do}(\Lambda=\lambda)} \mid X=x, Y=y, \hat{Y}=\hat{Y}'} \left[\mathbb{1}\{\hat{Y} \neq Y\}\right]\right] \\
&= \sum_{u,v} \sum_{\hat{y}} P(U = u, V = v \mid X = x, Y = y, \hat{Y} = \hat{y}) \cdot P^{\mathcal{M}}(\hat{Y} = \hat{y} \mid X = x, Y = y) \\
&\quad \cdot \mathbb{1}\{f_{\hat{Y}}(u, v, \mathcal{C}_\lambda(x)) \neq y\}
\end{aligned}
$$

and since $P(U = u, V = v \mid X = x, Y = y) = \sum_{\hat{y}} P(U = u, V = v \mid X = x, Y = y, \hat{Y} = \hat{y}) \cdot P^{\mathcal{M}}(\hat{Y} = \hat{y} \mid X = x, Y = y)$, we have

$$
\begin{aligned}
&\mathbb{E}_{\hat{Y}' \sim P^{\mathcal{M}}} \left[ \mathbb{E}_{\hat{Y} \sim P^{\mathcal{M};do(\Lambda = \lambda) \mid X = x, Y = y, \hat{Y} = \hat{Y}'}} \left[ \mathbb{1}\{\hat{Y} \neq Y\} \right] \right] \\
&= \sum_{u,v} P(U = u, V = v \mid X = x, Y = y) \cdot \mathbb{1}\{f_{\hat{Y}}(u, v, \mathcal{C}_\lambda(x)) \neq y\} \\
&= \sum_{u} P(U = u \mid X = x, Y = y) \cdot \sum_{v} P(V = v \mid X = x, Y = y) \cdot \mathbb{1}\{f_{\hat{Y}}(u, v, \mathcal{C}_\lambda(x)) \neq y\} \\
&= \mathbb{E}_{\hat{Y} \sim P^{\mathcal{M};do(\Lambda = \lambda)}} [\mathbb{1}\{\hat{Y} \neq Y\} \mid X = x, Y = y],
\end{aligned}
$$

where we used that the exogenous variables $U$ and $V$ are independent. ∎

**Lemma 3** *Under counterfactual monotonicity, for a given $x, y, \hat{y}$ the counterfactual harm $h_\lambda(x, y, \hat{y})$ is non-increasing as a function of $\lambda$ and right-continuous.*

*Proof.* If $y \in \mathcal{C}_\lambda(x) \ \forall \lambda \in \mathcal{L}$ then $h_\lambda(x, y, \hat{y}) = 0, \forall \lambda \in \Lambda$ given Eq. 8. If $\exists \tilde{\lambda} \in \Lambda$ such that $\tilde{\lambda} = \inf\{\lambda \in \Lambda : y \in \mathcal{C}_\lambda(x)\}$, then $\forall \lambda \geq \tilde{\lambda}$, it holds that $y \in \mathcal{C}_\lambda(x)$, and as a result $h_\lambda(x, y, \hat{y}) = 0$. For $\forall \lambda' < \tilde{\lambda}$, it holds that $y \notin \mathcal{C}_{\lambda'}(x)$ and as a result Eq. 8 becomes

$$
h_{\lambda'}(x, y, \hat{y}) = \mathbb{1}\{\hat{y} = y\} \cdot 1 = \mathbb{1}\{\hat{y} = y\} \geq 0 = h_\lambda(x, y, \hat{y}), \tag{25}
$$

where $\mathbb{1}\{\hat{y} = y\}$ is constant $\forall \lambda' < \tilde{\lambda}$. ∎

**Lemma 4** *For a given data sample $x$ with label $y$, let $G : \mathcal{L} \to \{0, 1\}$ be the function $G(\lambda) = \mathbb{1}\{\hat{Y} \neq y \wedge y \in \mathcal{C}_\lambda(x)\}$. Under interventional monotonicity, $G$ is non-decreasing in $\lambda$.*

*Proof.*

Let $\lambda, \lambda' \in \mathcal{L}$ such that $\lambda \leq \lambda'$. Then it will always hold that $\mathcal{C}_\lambda(x) \subseteq \mathcal{C}_{\lambda'}(x)$. We distinguish the following cases for $\mathcal{C}_\lambda(x), \mathcal{C}_{\lambda'}(x)$:

- $y \notin \mathcal{C}_{\lambda'}(x)$. Then $G(\lambda) = G(\lambda') = 0$.
- $y \notin \mathcal{C}_\lambda(x)$ and $y \in \mathcal{C}_{\lambda'}(x)$. Then, $G(\lambda) = 0$ and $G(\lambda') = \mathbb{1}\{\hat{Y} \neq y\} \geq 0 = G(\lambda)$
- $y \in \mathcal{C}_\lambda(x) \subseteq \mathcal{C}_{\lambda'}(x)$. Then, $G(\lambda) = G(\lambda') = \mathbb{1}\{\hat{Y} \neq y\}$.

∎

## C   Additional experimental details

In this section, we describe the experimental details that are omitted from the main paper due to space constraints.

**Mixture of multinomial logit models.** Multinomial logit models (MNLs) [60] are among the most popular discrete choice models used to predict the probability that a human chooses an alternative, given a set of alternatives within a particular *context* [61]. Straitouri et al. [17] used MNLs to predict the probability that a human expert predicts a label (*e.g.*, the ground truth label), given a prediction set, while considering as context the ground truth label and the level of difficulty of the image. To this end, for images with the same level of difficulty, they estimate the confusion matrix of the expert predictions on their own and use it to parameterize an MNL model. Therefore, their mixture of MNLs comprises of a different MNL model for each level of difficulty. To distinguish images based on their level of difficulty, they consider different quantiles of the experts' average accuracy values of all images. In our experiments, we consider two levels of difficulty, where in the first level we consider images for which the average accuracy of experts is smaller than the $0.5$ quantile, and consider the rest of the images in the second level.

**Labels of the ImageNet16H dataset.** The ImageNet16H [52] was created using $1{,}200$ unique images labeled into $207$ different fine-grained categories from the ILSRVR 2012 dataset [53]. Steyvers et al. [52] mapped each of these fine-grained categories into one out of $16$ coarse-grained categories—namely airplane, bear, bicycle, bird, boat, bottle, car, cat, chair, clock, dog, elephant, keyboard, knife, oven, and truck—that serve as the ground truth labels. This mapping essentially eliminated any label disagreement that could potentially occur between annotators in the ILSRVR 2012 dataset.

**Implementation details and licenses.** We implement our methods and execute our experiments using `Python` $3.10.9$, along with the open-source libraries `NumPy` `1.26.4` (BSD License), and `Pandas` `2.2.1` (BSD 3-Clause License). For reproducibility, we used a different fixed random seed for each random sampling of the test and calibration sets. Both datasets used in our experiments are distributed under the Creative Commons Attribution License 4.0 (CC BY 4.0).

# D  Additional experimental results on counterfactual monotonicity

In this section, we evaluate our methodology on additional strata of images and on larger calibration sets, and study the relationship between average counterfactual harm, average prediction set size and empirical coverage.

**Other strata of images.** Figures 4 and 5 show the average accuracy $A(\lambda)$ achieved by a human expert using $\mathcal{C}_\lambda$, as predicted by the mixture of MNLs, against the average counterfactual harm $H(\lambda)$ caused by $\mathcal{C}_\lambda$ for $\alpha \in \{0.01, 0.05\}$ on the strata of images with $\omega = 80$ and $\omega = 95$, respectively. The meaning and coloring of each point are the same as in Figure 2. The results on both strata of images are consistent with the results on the stratum of images with $\omega = 110$—the decision support systems $\mathcal{C}_\lambda$ always cause some amount of counterfactual harm, our framework successfully identifies harm-controlling sets and there is a trade-off between accuracy and counterfactual harm. However, for the strata with $\omega \in \{80, 95\}$, where the classification task has lower difficulty compared to the stratum with $\omega = 110$, we observe that the trade-off between accuracy and counterfactual is minimal.

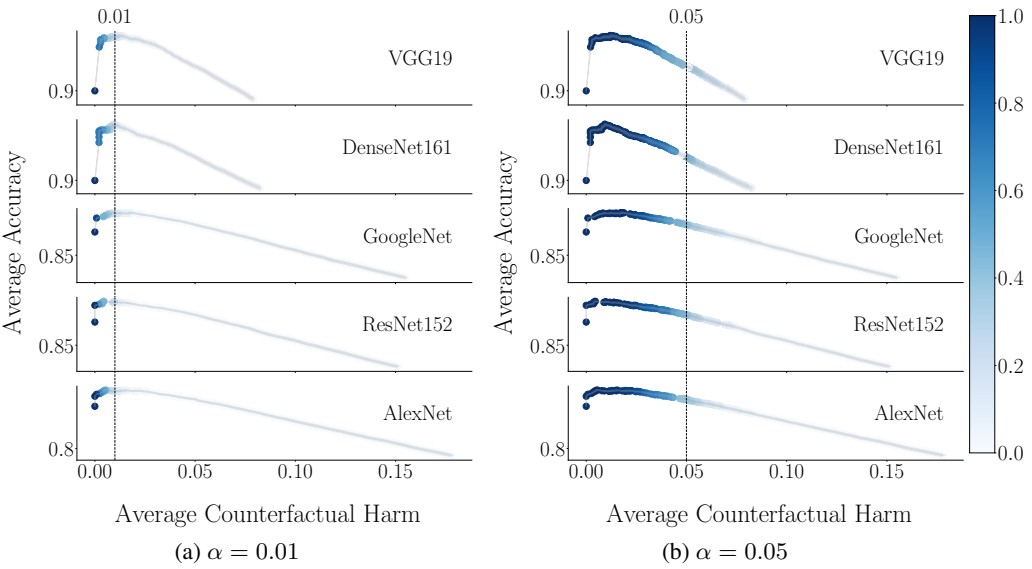

(a) $\alpha = 0.01$          (b) $\alpha = 0.05$

Figure 4: Average accuracy estimated by the mixture of MNLs against the average counterfactual harm for images with $\omega = 80$. Each point corresponds to a $\lambda$ value from $0$ to $1$ with step $0.001$ and the coloring indicates the relative frequency with which each $\lambda$ value is in $\Lambda(\alpha)$ across random samplings of the calibration set. Each row corresponds to decision support systems $\mathcal{C}_\lambda$ with a different pre-trained classifier with average accuracies $0.891$ (VGG19), $0.892$ (DenseNet161), $0.802$ (GoogleNet), $0.804$ (ResNet152), and $0.784$ (AlexNet). The average accuracy achieved by human experts on their own is $0.9$. The results are averaged across $50$ random samplings of the test and calibration set. In both panels, $95\%$ confidence intervals have width always below $0.02$ and are represented using shaded areas.

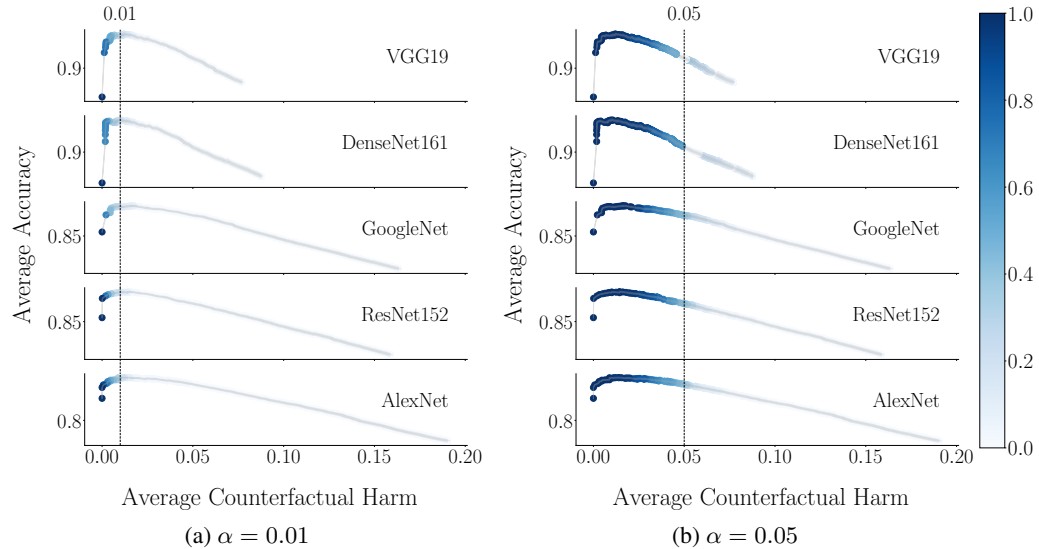

(a) $\alpha = 0.01$          (b) $\alpha = 0.05$

Figure 5: Average accuracy estimated by the mixture of MNLs against the average counterfactual harm for images with $\omega = 95$. Each point corresponds to a $\lambda$ value from 0 to 1 with step 0.001 and the coloring indicates the relative frequency with which each $\lambda$ value is in $\Lambda(\alpha)$ across random samplings of the calibration set. Each row corresponds to decision support systems $\mathcal{C}_\lambda$ with a different pre-trained classifier with average accuracies 0.88 (VGG19), 0.868 (DenseNet161), 0.775 (GoogleNet), 0.773 (ResNet152), and 0.745 (AlexNet). The average accuracy achieved by human experts on their own is 0.86. The results are averaged across 50 random samplings of the test and calibration set. In both panels, 95% confidence intervals have width always below 0.02 and are represented using shaded areas.

**Larger calibration sets.** Figure 6 shows the average accuracy $A(\lambda)$ achieved by a human expert using $\mathcal{C}_\lambda$, as predicted by the mixture of MNLs, against the average counterfactual harm $H(\lambda)$ caused by $\mathcal{C}_\lambda$ for $\alpha = 0.01$ on the stratum of images with $\omega = 110$. The results show that the larger the calibration set, the more often $\lambda$ values with average counterfactual harm $H(\lambda)$ close to $\alpha$[12] are included in the harm-controlling set $\Lambda(\alpha)$. This is because, the larger the number of samples in the calibration set ($n$), the lower the estimation error in the empirical estimate of the average counterfactual harm $\hat{H}_n$. We find qualitatively similar results for the other strata of images.

**Average prediction set-size and empirical coverage.** Figure 7(a) shows the average prediction set-size of each decision support system $\mathcal{C}_\lambda$ against the average counterfactual harm $H(\lambda)$ caused by $\mathcal{C}_\lambda$ on the stratum of images with $\omega = 110$. Figure 7(b) shows the empirical coverage, *i.e.*, the fraction of samples $x$ in the test set, for which ground truth label $y \in \mathcal{C}_\lambda(x)$, for each decision support system $\mathcal{C}_\lambda$ against the average counterfactual harm $H(\lambda)$ caused by $\mathcal{C}_\lambda$ on the same stratum of images. The results show that systems with higher coverage construct larger prediction sets in average and cause less counterfactual harm. We find similar results for the other strata of images.

---

[12]Here, note that for $\alpha = 0.01$, these $\lambda$ values happen to achieve the highest average accuracy $A(\lambda)$.

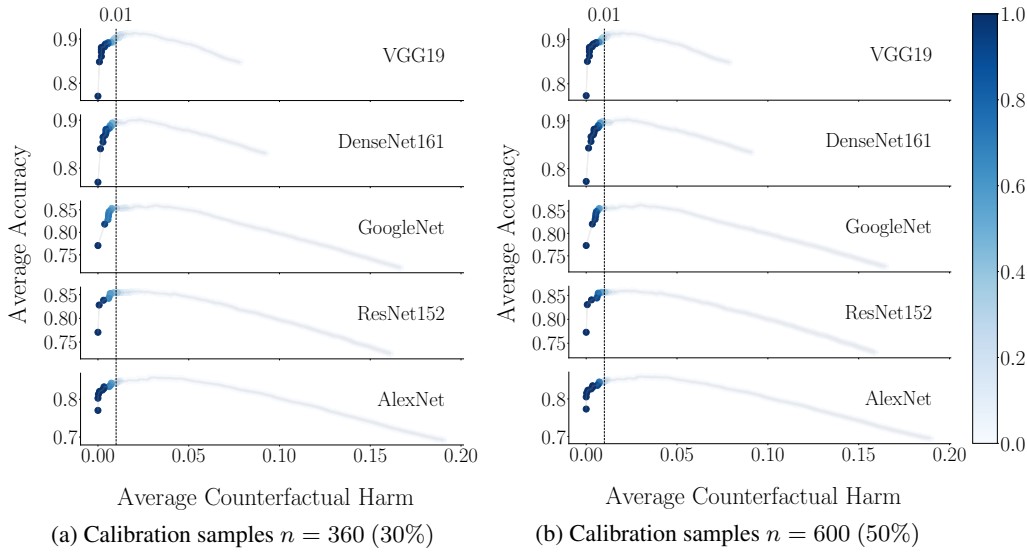

(a) Calibration samples $n = 360$ (30%)   (b) Calibration samples $n = 600$ (50%)

Figure 6: Average accuracy estimated by the mixture of MNLs against the average counterfactual harm for images with $\omega = 110$ and different sizes of the calibration set. Each point corresponds to a $\lambda$ value from 0 to 1 with step 0.001 and the coloring indicates the relative frequency with which each $\lambda$ value is in $\Lambda(\alpha)$ across random samplings of the calibration set. Each row corresponds to decision support systems $\mathcal{C}_\lambda$ with a different pre-trained classifier with average accuracies 0.846 (VGG19), 0.830 (DenseNet), 0.722 (GoogleNet), 0.727 (ResNet152), and 0.691 (AlexNet). The average accuracy achieved by the simulated human experts on their own is 0.771. The results are averaged across 50 random samplings of the test and calibration set. In both panels, 95% confidence intervals are represented using shaded areas and always have width below 0.02.

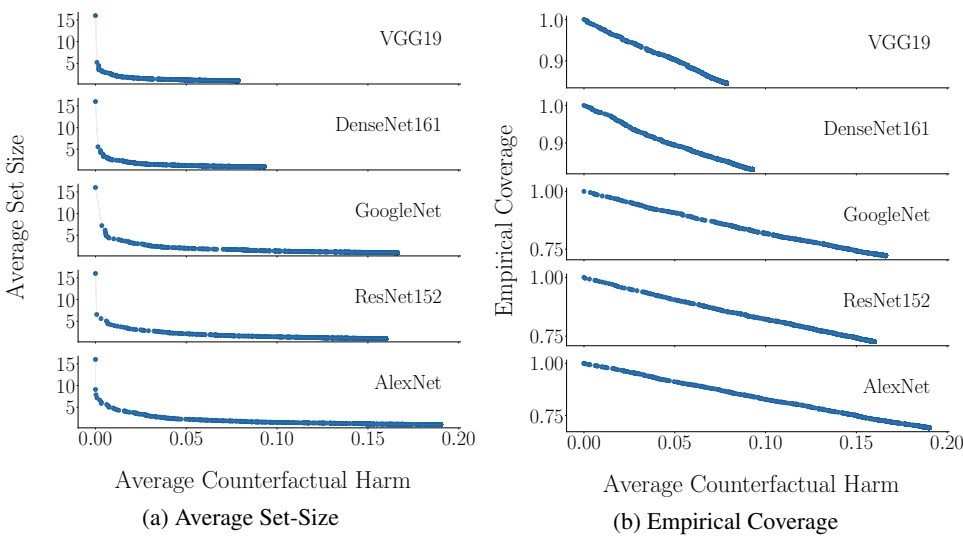

(a) Average Set-Size   (b) Empirical Coverage

Figure 7: Average set size and empirical coverage against the average counterfactual harm for images with $\omega = 110$. Each point corresponds to a $\lambda$ value from 0 to 1 with step 0.001. Each row corresponds to decision support systems $\mathcal{C}_\lambda$ with a different pre-trained classifier with average accuracies as in Figure 6 above. The average accuracy achieved by the simulated human experts on their own is 0.771. The results are averaged across 50 random samplings of the test and calibration set. In both panels, 95% confidence intervals are represented using shaded areas and always have width below 0.025.

# E Experimental results on counterfactual monotonicity with a more complex set-valued predictor

In this section, we experiment with a system that constructs the prediction set $\mathcal{C}(x)$ using a more complex set-valued predictor [49]. This set-valued predictor first computes the following nonconformity score for each label value $y \in \mathcal{Y}$ using the softmax output of a pre-trained classifier $m_y(x) \in [0, 1]$:

$$s(y, x, u; w) = \begin{cases} u \cdot m_{y_{(1)}}(x), & \text{if } y = y_{(1)} \\ m_{y_{(1)}} + (i - 2 + u) \cdot w, & \text{if } y = y_{(i)} \end{cases}, \tag{26}$$

where $y_{(i)}$ denotes the label value with the $i$-th largest classifier output, $u \sim U(0, 1)$ is a uniform random variable and $w$ is a hyperparameter representing the weight of the ranking of the label value. Then, given a user-specified threshold $\lambda \in \mathbb{R}$, it constructs the prediction sets $\mathcal{C}(x) = \mathcal{C}_\lambda(x)$ as follows:

$$\mathcal{C}_\lambda(x) = \{y_{(L-i)}\}_{i=0}^{k-1}, \text{ with } k = 1 + \sum_{j=1}^{L-1} \mathbb{1}\{s(y_{(j)}, x, u; w) \leq \lambda\}, \tag{27}$$

where, with a slight abuse of notation, $y_{(i)}$ is the label value with the $i$-th largest $s(y, x, u; w)$ value. Here, we should note that may exist data samples with $s(y, x, u; w) > 1$. Therefore, considering $\lambda \in [0, 1]$ may not include every set-valued predictor given by Eq. 27. For this reason, in what follows, we consider $\lambda \in [0, \max s(y, x, u, w)]$, where the maximum is essentially over the classifier output $m_y(x)$, the uniform random variable $u$ and the hyperparameter $w$.

**Experimental setup.** We use the ImageNet16H dataset and follow the same setup as described in Section 6, with only difference that we split the images into a calibration set (10%), which we use to find $\Lambda(\alpha)$, a validation set (10%), which we use to select the value of the hyperparameter $w$ following a similar procedure as in Huang et al. [49], and a test set (80%). In each iteration of the experiment, given a $\lambda$ value, we select the value of $w \in \{0.02, 0.05, 0.1, 0.15, 0.2, 0.25, 0.3, 0.35\}$ for which the set-valued predictor achieves the smallest average prediction set-size over the data samples in the validation set.

**Results.** The meaning and coloring of each point are the same as in Figure 2. The results on all strata of images are consistent with the results using the set-valued predictor given by Eq. 1—the decision support systems $\mathcal{C}_\lambda$ always cause some amount of counterfactual harm, our framework successfully identifies harm-controlling sets and there is a trade-off between accuracy and counterfactual harm.

Figure 11(a) shows the average prediction set-size of each decision support system $\mathcal{C}_\lambda$ against the average counterfactual harm $H(\lambda)$ caused by $\mathcal{C}_\lambda$ on the stratum of images with $\omega = 110$. Figure 11(b) shows the empirical coverage for each decision support system $\mathcal{C}_\lambda$ against the average counterfactual harm $H(\lambda)$ caused by $\mathcal{C}_\lambda$ on the stratum of images with $\omega = 110$. Our results are consistent with the results for the set-valued predictor given by Eq. 1 in Figure 7—systems $\mathcal{C}_\lambda$ that achieve higher coverage, construct larger prediction sets on average and cause less harm.

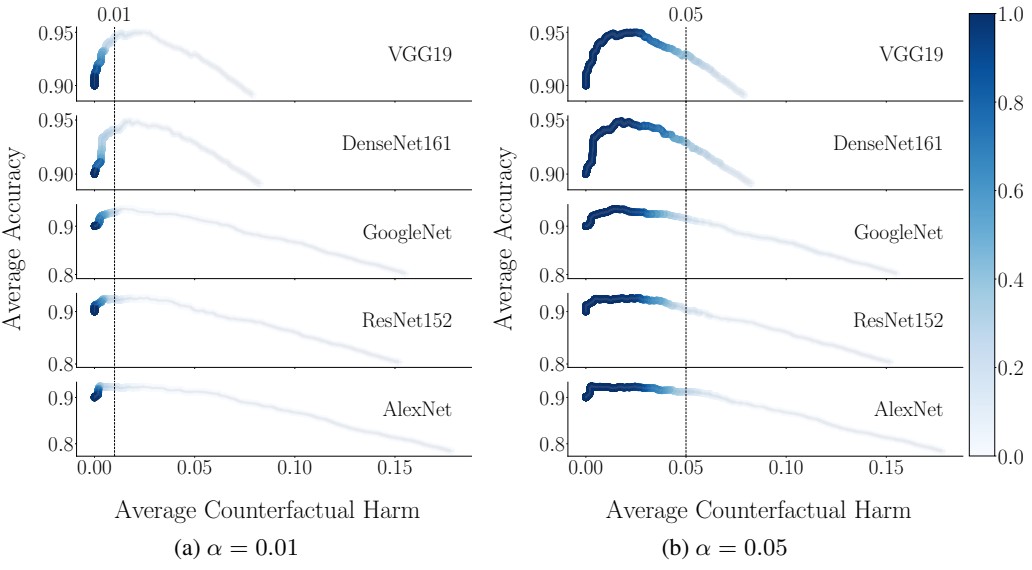

(a) $\alpha = 0.01$               (b) $\alpha = 0.05$

Figure 8: Average accuracy estimated by the mixture of MNLs against the average counterfactual harm for images with $\omega = 80$. Each point corresponds to a $\lambda$ value from 0 to 6.25 with step 0.00625 and the coloring indicates the relative frequency with which each $\lambda$ value is in $\Lambda(\alpha)$ across random samplings of the calibration set. Each row corresponds to decision support systems $\mathcal{C}_\lambda$ with a different pre-trained classifier with average accuracies 0.891 (VGG19), 0.892 (DenseNet161), 0.802 (GoogleNet), 0.804 (ResNet152), and 0.784 (AlexNet). The average accuracy of human experts on their own is 0.9. The results are averaged across 50 random samplings of the test and calibration set. In both panels, 95% confidence intervals have width always below 0.02 and are represented using shaded areas.

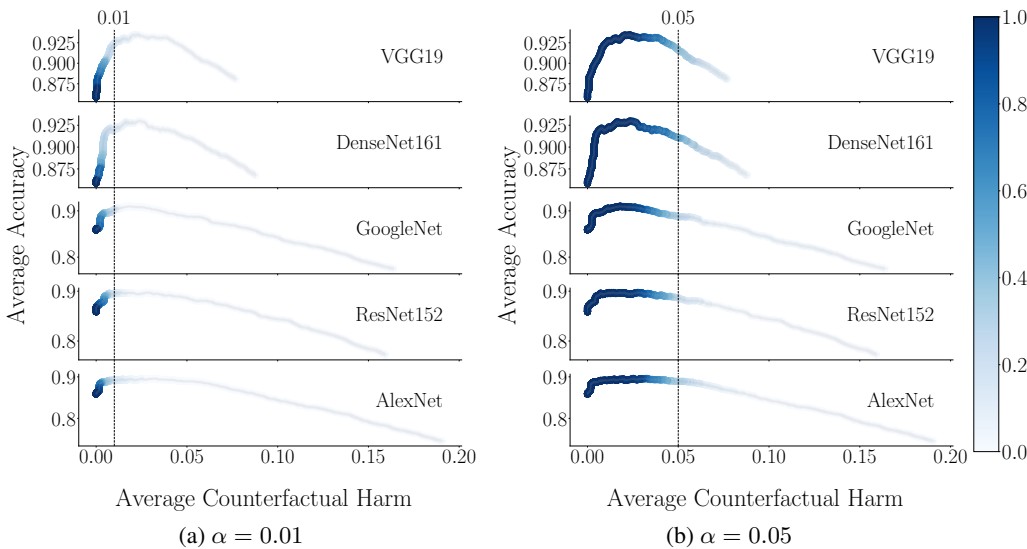

(a) $\alpha = 0.01$               (b) $\alpha = 0.05$

Figure 9: Average accuracy estimated by the mixture of MNLs against the average counterfactual harm for images with $\omega = 95$. Each point corresponds to a $\lambda$ value from 0 to 6.25 with step 0.00625 and the coloring indicates the relative frequency with which each $\lambda$ value is in $\Lambda(\alpha)$ across random samplings of the calibration set. Each row corresponds to decision support systems $\mathcal{C}_\lambda$ with a different pre-trained classifier with average accuracies 0.88 (VGG19), 0.868 (DenseNet161), 0.775 (GoogleNet), 0.773 (ResNet152), and 0.745 (AlexNet). The average accuracy of human experts on their own is 0.86. The results are averaged across 50 random samplings of the test and calibration set. In both panels, 95% confidence intervals have width always below 0.02 and are represented using shaded areas.

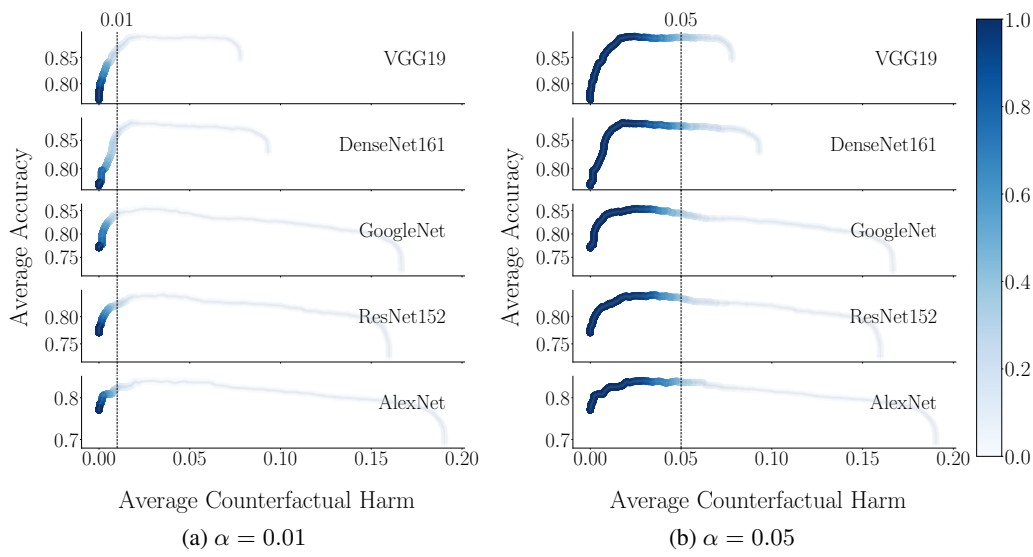

(a) $\alpha = 0.01$
(b) $\alpha = 0.05$

Figure 10: Average accuracy estimated by the mixture of MNLs against the average counterfactual harm for images with $\omega = 110$. Each point corresponds to a $\lambda$ value from 0 to 6.25 with step 0.00625 and the coloring indicates the relative frequency with which each $\lambda$ value is in $\Lambda(\alpha)$ across random samplings of the calibration set. Each row corresponds to decision support systems $\mathcal{C}_\lambda$ with a different pre-trained classifier with average accuracies 0.846 (VGG19), 0.830 (DenseNet), 0.722 (GoogleNet), 0.727 (ResNet152), and 0.691 (AlexNet). The average accuracy achieved by the simulated human experts on their own is 0.771. The results are averaged across 50 random samplings of the test and calibration set. In both panels, 95% confidence intervals are represented using shaded areas and always have width below 0.02.

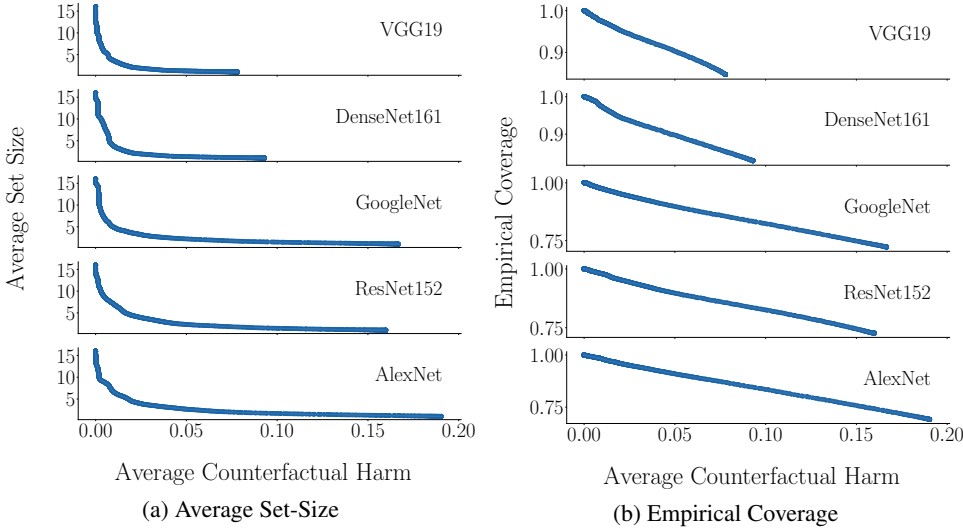

(a) Average Set-Size
(b) Empirical Coverage

Figure 11: Average set size and empirical coverage against the average counterfactual harm for images with $\omega = 110$. Each point corresponds to a $\lambda$ value from 0 to 6.25 with step 0.00625. Each row corresponds to decision support systems $\mathcal{C}_\lambda$ with a different pre-trained classifier with average accuracies as in Figure 10 above. The average accuracy achieved by the simulated human experts on their own is 0.771. The results are averaged across 50 random samplings of the test and calibration set. In both panels, 95% confidence intervals are represented using shaded areas and always have width below 0.025.

# F  Average accuracy vs. prediction set size

In this section, we use the ImageNet16H-PS [18] dataset to verify the interventional monotonicity assumption, *i.e.*, whether human experts achieve higher average accuracy when predicting from smaller prediction sets that include the ground truth label. To this end, we follow a procedure, similar to the one used by Straitouri et al. [18] to verify the interventional monotonicity assumption unconditionally of the value of the ground truth label.

Straitouri et al. estimate the average accuracy per prediction set size on images with similar difficulty, averaged across all experts and across experts with the same level of competence. They consider images with similar difficulty instead of single images, as the dataset ImageNet16H-PS does not include enough human predictions per image to faithfully estimate the average accuracy per image. They stratify the images into groups of similar difficulty, following the same procedure used in Straitouri et al. [17] based on different quantiles of the average accuracy values of all images. In our work, we stratify images with the same ground truth label into the following four groups:

— High difficulty: images with average accuracy within the $0.25$ quantile of the average accuracy values of all images with the same ground truth label.

— Medium to high difficulty: images with average accuracy within the $0.5$ quantile and outside the $0.25$ quantile of the average accuracy values of all images with the same ground truth label.

— Medium to low difficulty: images with average accuracy within the $0.75$ quantile and outside the $0.5$ quantile the of the average accuracy values of all images with the same ground truth label.

— Low difficulty: images with average accuracy outside the $0.75$ quantile of the average accuracy values of all images with the same ground truth label.

We follow a similar method to stratify experts based on their level of competence into two groups for each ground truth label. To measure the level of competence of an expert for a ground truth label, we use the average accuracy across all the predictions that she made on images with this ground truth label. For each ground truth label, we consider the $50\%$ of experts with the highest average accuracy as the experts with high level of competence and the rest $50\%$ as experts with low level of competence.

Figure 12 shows the average accuracy per prediction set size—for prediction sets that include the ground truth label—across images of various levels of difficulty and experts with different levels of competence for some of the ground truth labels. We find qualitatively similar results for the rest of the ground truth labels. The results are consistent with the results on the unconditional interventional monotonicity by Straitouri et al. [18], showing that as long as the classification task is not too easy, the experts achieve higher average accuracy when predicting from smaller prediction sets, *i.e.*, the interventional monotonicity holds.

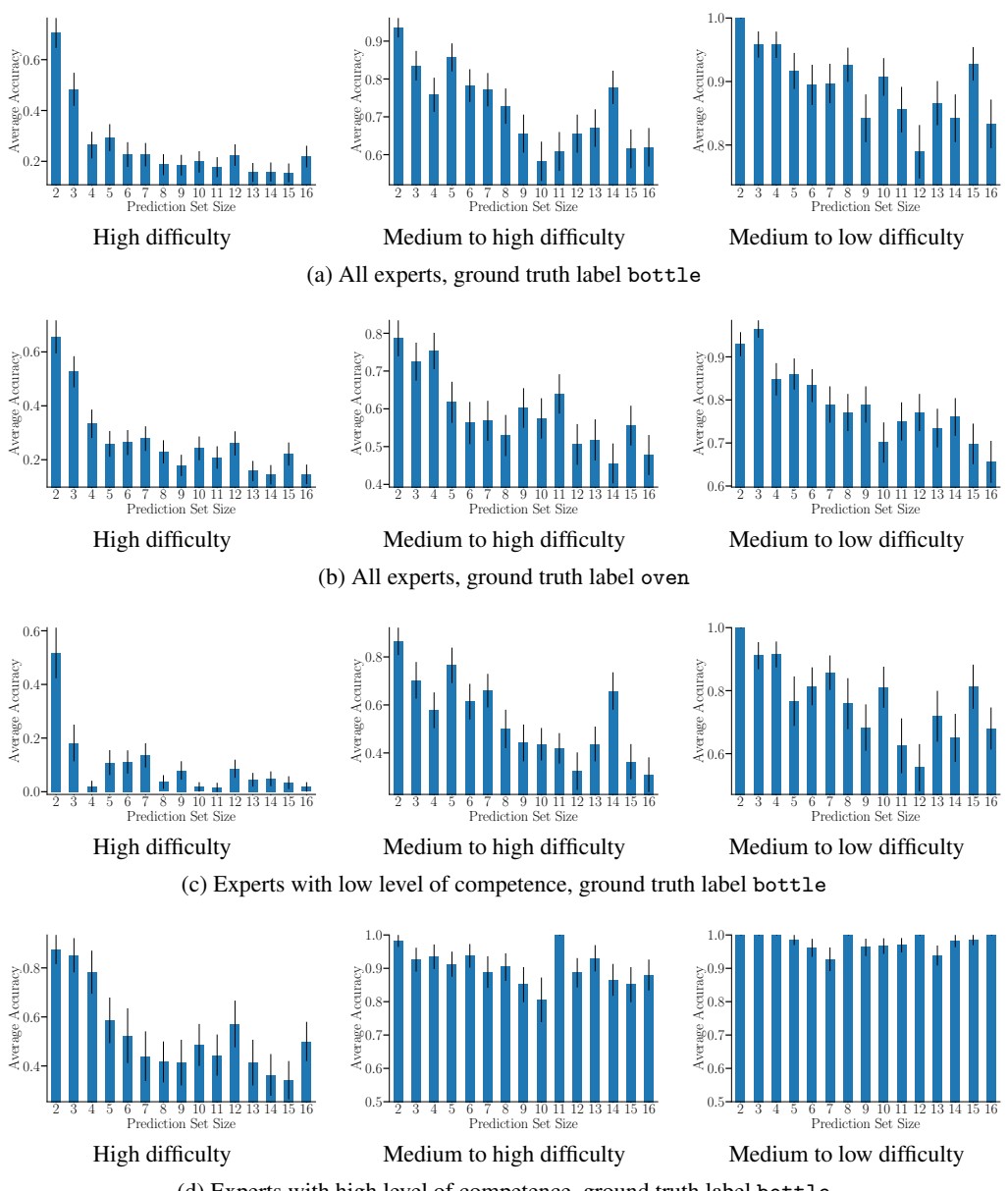

(a) All experts, ground truth label `bottle`

(b) All experts, ground truth label `oven`

(c) Experts with low level of competence, ground truth label `bottle`

(d) Experts with high level of competence, ground truth label `bottle`

Figure 12: Empirical success probability per prediction set size averaged across (a) all experts for images with ground truth label `bottle`, (b) all experts for images with ground truth label `oven`, (c) experts with low level of competence for images with ground truth label `bottle`, and (d) experts with high level of competence for images with ground truth label `bottle` with high, medium to high and medium to low difficulty. In all panels, we have only considered prediction sets that included the ground truth label and thus have omitted showing the empirical success probability for singletons, as it is always 1. Error bars denote standard error.

# G   Experiments under the interventional monotonicity assumption

In this section, we show the average accuracy $A(\lambda)$ achieved by a human expert using $\mathcal{C}_\lambda$, as predicted by the mixture of MNLs, against the average counterfactual harm upper-bound caused by $\mathcal{C}_\lambda$ for several $\alpha$ values on the strata of images with $\omega = 80$ in Figure 13, with $\omega = 95$ in Figure 14 and with $\omega = 110$ in Figure 15. Here, each point corresponds to a different $\lambda$ value and its coloring indicates the empirical probability that $\lambda$ is included in the harm-controlling set $\Lambda'(\alpha)$ given by Corollary 2, for a fixed $\alpha'$ value across the random samplings of the test and calibration sets. For each classifier and stratum of images, we select the $\alpha'$ value maximizing the expected size of the harm-controlling set $\Lambda'(\alpha)$. The results on all strata of images support the findings derived from the experiments assuming that the counterfactual monotonicity assumption holds. In addition, we observe that the gap between the lower-bound of the average counterfactual harm (that is equal to the average counterfactual harm under the counterfactual monotonicity assumption) and the upper-bound is often large.

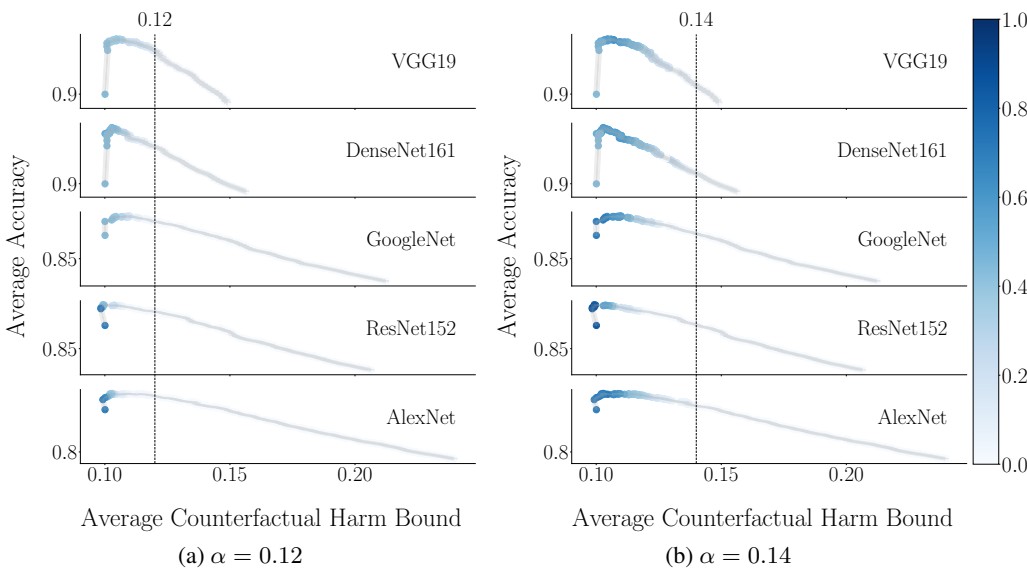

Figure 13: Average accuracy estimated by the mixture of MNLs against the average counterfactual harm upper bound for images with $\omega = 80$. Each point corresponds to a $\lambda$ value from 0 to 1 with step 0.001 and the coloring indicates the relative frequency with which each $\lambda$ value is in $\Lambda'(\alpha)$ across random samplings of the calibration set for a fixed $\alpha'$. Each row corresponds to decision support systems with a different pre-trained classifiers as in Figure 4. The results are averaged across 50 random samplings of the test and calibration set. In both panels, $95\%$ confidence intervals have width always below 0.02 and are represented using shaded areas.

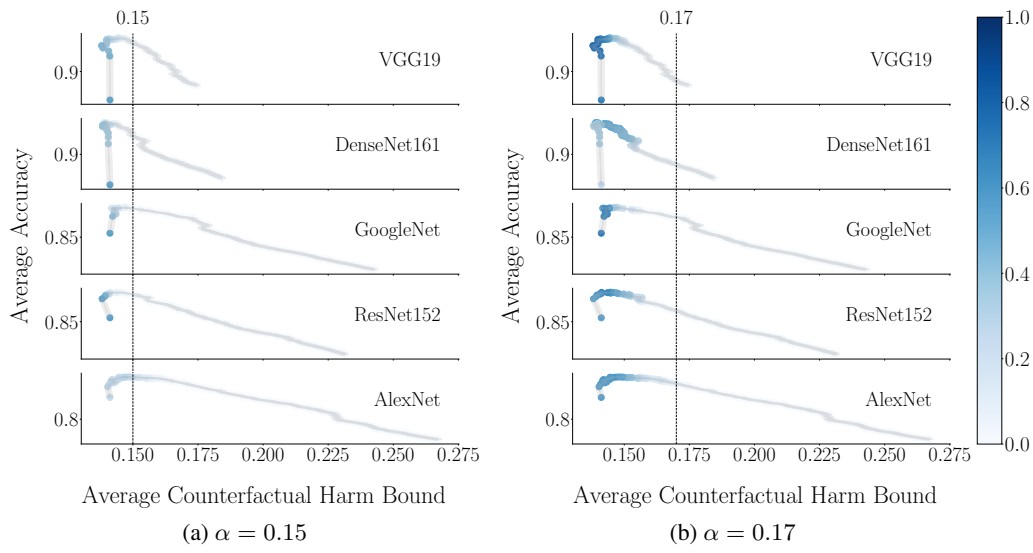

Figure 14: Average accuracy estimated by the mixture of MNLs against the average counterfactual harm upper bound for images with $\omega = 95$. Each point corresponds to a $\lambda$ value from 0 to 1 with step 0.001 and the coloring indicates the relative frequency with which each $\lambda$ value is in $\Lambda'(\alpha)$ across random samplings of the calibration set for a fixed $\alpha'$. Each row corresponds to decision support systems with a different pre-trained classifiers as in Figure 5. The results are averaged across 50 random samplings of the test and calibration set. In both panels, 95% confidence intervals have width always below 0.02 and are represented using shaded areas.

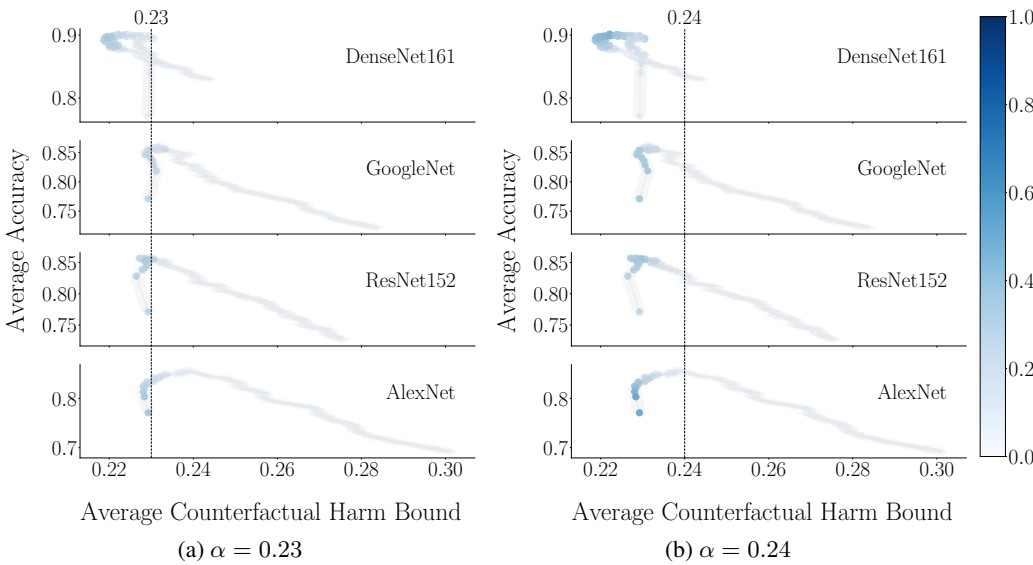

Figure 15: Average accuracy estimated by the mixture of MNLs against the average counterfactual harm upper bound for images with $\omega = 110$. Each point corresponds to a $\lambda$ value from 0 to 1 with step 0.001 and the coloring indicates the relative frequency with which each $\lambda$ value is in $\Lambda'(\alpha)$ across random samplings of the calibration set for a fixed $\alpha'$. Each row corresponds to decision support systems with a different pre-trained classifiers as in Figure 2. For the classifier VGG19, the average counterfactual harm bound is $\sim 0.23$ for each $\lambda$ value. The results are averaged across 50 random samplings of the test and calibration set. In both panels, 95% confidence intervals have width always below 0.02 and are represented using shaded areas.

