# OpenReview forum: "Controlling Counterfactual Harm in Decision Support Systems Based on Prediction Sets"
_NeurIPS.cc/2024/Conference — NeurIPS 2024 poster_

### Official Review · Reviewer_KmhA · 2024-06-12

**Soundness:** 4
**Presentation:** 3
**Contribution:** 3
**Rating:** 6
**Confidence:** 4

**Summary:**

The paper introduces a notion of counterfactual harm for a classification setting in which a human is choosing among a subset of possible labels determined by an AI system, i.e., the automated system is used to "narrow down" the set of most likely labels for a sample. Under assumptions of counterfactual and interventional monotonicity, which imply that , the notion of harm can be identified pointwise or up to an interval. A procedure for controlling harm is introduced, based on conformal risk control.

**Strengths:**

(S1) The described problem is important. Furthermore, not many works in this space exist, and the definition of harm is not universally agreed upon, as noted by the authors. Thus, this paper explores a valuable direction of research.

(S2) Many of the concepts are introduced in a clean way, and the writing is good overall.

(S3) The paper seems to combine many state-of-the-art ideas, including notions of counterfactual harm, identification under monotonicity assumptions from previous works, and conformal risk control.

**Weaknesses:**

Major:

(W1) The causal inference notation is not up to the expected standard. In Definition 1, which is key to the paper, the notation does not work. The paper is trying to describe the notion of counterfactual harm, while using only the $do()$ operator and conditioning. The do() operator and conditioning can only give quantities in the first two layers of the causal hierarchy. Therefore, the current definition, as written, is incorrect. The appearance of $\hat Y$ in multiple places with different meanings is concerning. Instead of this, the paper probably wants to use the subscript notation $\hat Y _{\lambda}$.

My understanding is that the paper is trying to say

$$ h_\lambda (x, y, \hat y) = E [ max (0, \mathbb{1} ( \hat Y = Y ) - \mathbb{1} (\hat Y _{\lambda} = Y) ) \mid X = x, \hat Y = \hat y, Y = y]. $$

(W2) The connection to Richens et. al [1] is mentioned in the introduction. It is clarified that the difference here is that "their definition requires one to deploy the system", while this is not the case for the definition proposed in this work. It is not clear from the text whether  the current definition does not require deployment because of the definition itself, or the monotonicity assumptions that are invoked.
The two definitions seem closely related, and given the comment in (W1), it would be very valuable to have these two definitions compared side-by-side, with a unified choice of corrected notation. Opening a small appendix for this would be very desirable.

---

Minor comments / corrections:

(W3) Relating to point (W1): in Pearl's notation, causal mechanisms are usually denoted by the assignment operator $\gets$, i.e., $X \gets f_X (U_X)$. This is useful to indicate that the mechanisms are not "equations that can be manipulated".

(C1) Line 159: "if they has failed"

**Questions:**

Please see (W1), (W2).

(Q1) Lines 147-148 claim that the $\Lambda (\alpha)$ set cannot be found because counterfactual harm is a counterfactual quantity. This statement seems imprecise. For instance, the effect of treatment on the treated (ETT), is a counterfactual quantity that can be estimated with lack-of-confounding assumptions. Furthermore, any counterfactual quantity could be, in principle, bounded without additional assumptions.

It seems more likely that the joint distribution of $\hat Y, \hat Y _{\lambda}$ is required to evaluate the expression (and this seems to be a consequence of the max() operator within the expectation), which is known hard problem, often solved with monotonicity assumptions.

(Q2) When taking a step back, the question that comes to mind is: what is the end goal of the proposal? When minimizing counterfactual harm, it seems that "counterfactual benefit" is almost entirely ignored. Is this a reasonable notion to ignore? Furthermore, if the definition were to remove the max(), would this be equivalent to optimizing overall accuracy or something similar (note that removing the max solves the identifiability issue)? Finally, should there be an extension in which $\lambda = \lambda(x)$ is a function of $X = x$, so that harm is minimized in areas where it is largest?

(Q3) Line 142: "Otherwise, the expert's prediction could not have become worse". The claim of the paper is about decision-making support systems, whereas this comment seems quite closely focused on accuracy. For instance, in the very example given in Figure 1: if someone has a common cold, mistaking this for a flu seems rather different than mistaking it for lung cancer (the consequences of these two classifications may be rather different)? Please comment on this.

(Q4) Lines 127-130 are saying that an SCM can match the observational distribution. Can you comment on the significance of this observtion? Is this remark made to claim that the usage of SCMs does not limit the generality of the approach, due to their expressiveness? If so, perhaps it is worth adding such a sentence.

**Limitations:**

Please check (W1), (W2), and (Q1), (Q3).

---

> ### Author Rebuttal · Authors · 2024-08-05
>
> **[W1]** In our paper, we follow the same notation used in Elements of Causal Inference by Peters, Janzing and Schölkopf, a standard book in causality research with over 2,000 citations in Google Scholar, also used in [18] (ICML 24). In particular, under the notation used by Peters et al. to denote interventions in a counterfactual distribution in Chapter 3.3, we think our current definition is correct. In Definition 1, the distribution $P^{\mathcal{M} ; do(\Lambda = \lambda) | \hat{Y} = \hat{y}, X = x, Y=y}(\hat{Y})$ denotes the counterfactual distribution entailed by the structural causal model $\mathcal{M}$ under the intervention $do(\Lambda = \lambda)$ given the observation $\hat{Y} = \hat{y}, X = x, Y=y$. We would prefer to keep the notation as is because it explicitly notes the (counterfactual) distribution used in the expectation. However, to avoid misunderstandings, we will describe our notation in the revised paper.
>
> **[W2]** In the notation we use, following Peters et al. as noted above, the definition by Richens et al. is as follows: $$h_{\lambda}(x, y, \hat{y}) = E\_{\hat{Y} \sim P^{\mathcal{M} | \hat{Y} = \hat{y}, X = x, Y=y}(\hat{Y})}[ \max\\{0, \Bbb{1}\\{\hat{Y} = y\\} - \Bbb{1}\\{\hat{y} = y\\}\\} ],$$ where $x, y, \hat{y} \sim P^{\mathcal{M} ; do(\Lambda=\lambda)}$ and we consider $\lambda=1$ to be the default action (in Richen's language). In contrast, our definition is as follows: $$h_{\lambda}(x, y, \hat{y}) = E_{\hat{Y} \sim P^{\mathcal{M} ; do(\Lambda = \lambda) | \hat{Y} = \hat{y}, X = x, Y=y}(\hat{Y})}[ \max\\{0, \Bbb{1}\\{\hat{y} = y\\}  - \Bbb{1}\\{\hat{Y} = y\\} \\} ],$$ where $x, y, \hat{y} \sim P^{\mathcal{M}}$. Under Richen's definition, counterfactual harm depends on an observation $x, y, \hat{y} \sim P^{\mathcal{M} ; do(\Lambda=\lambda)}$, which comprises a setting in which the human expert makes a prediction from the prediction set $\mathcal{C}\_{\lambda}(x)$ (i.e, the decision support system $\mathcal{C}_{\lambda}$ is deployed). In contrast, under our definition, counterfactual harm depends on an observation $x, y, \hat{y} \sim P^{\mathcal{M}}$, which comprises a setting in which the human expert makes a prediction on their own.
>
> In the notation suggested by the reviewer, the definition by Richens et al. is as follows: $$h_{\lambda}(x, y, \hat{y}) = \mathbb{E}[\max\\{0, \Bbb{1}\\{\hat{Y}\_{1} = y \\} - \Bbb{1}\\{\hat{y} = y \\} \\} | X=x, Y=y, \hat{Y}\_{\lambda}=\hat{y}],$$ where $x, y, \hat{y} \sim P(X, Y, \hat{Y}\_{\lambda})$ and the expectation is over the prediction $\hat{Y}\_{1}$ made by the expert on their own and we consider $\lambda=1$ to be the default action. In contrast, our definition is as follows: $$h_{\lambda}(x, y, \hat{y}) = \mathbb{E}[\max\\{0, \Bbb{1}\\{\hat{y} = y \\} - \Bbb{1}\\{\hat{Y}\_{\lambda} = y\\} \\} | X=x, Y=y, \hat{Y}\_{1}=\hat{y}],$$ where $x, y, \hat{y} \sim P(X, Y, \hat{Y}\_{1})$ and the expectation is over the prediction $\hat{Y}\_{\lambda}$ made by the expert using the system $\mathcal{C}\_{\lambda}$.
>
> Following the reviewer's advice, we will clarify this in a separate Appendix.
>
> **[W3]** As noted above, we follow standard notation in causality research, which we would respectfully prefer to keep, though, we will explicitly clarify that the mechanisms are not equations that can be manipulated in the revised paper.
>
> **[Q1]** To avoid misunderstandings, we will rephrase lines 147-148 to clarify that we cannot expect to find  $\Lambda (\alpha)$ from observational data as counterfactual harm is not identifiable from observational data without any further assumptions. However, we would like to clarify that, since $\hat{Y}$ and $\hat{Y}_{\lambda}$ are the same variable under no intervention $(\Lambda=1)$ and under the intervention $do(\Lambda=\lambda)$, it is not possible to simultaneously measure them, therefore it is not sensible to consider their joint distribution as argued in Pearl, J. (2010). An introduction to causal inference. The international journal of biostatistics, 6(2) (page 25).
>
> **[Q2]** We would like to clarify that our goal is not to minimize counterfactual harm but rather to find the largest harm-controlling set $\Lambda(\alpha)$ for a user-specified harm bound $\alpha$, as described in lines 144-150. Among the $\lambda$ values in  $\Lambda(\alpha)$, one could later on find the $\lambda$ value that maximizes some utility, e.g., accuracy by using the methods in [17,18], or the counterfactual benefit, though this remains outside the scope of our work.
>
> Removing the max from our definition of counterfactual harm would result in measuring the accuracy of the human expert under no interventions minus the accuracy of their counterfactual predictions had they used the system. That said, removing the max would not solve the identifiability issue, as this is due to the counterfactual predictions in the definition.
>
> To the best of our knowledge, we have found no prior work on conformal risk control using $\lambda = \lambda(x)$ to construct the prediction sets, however, a recent work by Gibbs et al. (Conformal Prediction With Conditional Guarantees, Arxiv 2023) uses threshold values that depend on x in the context of conformal prediction. We will add this in our discussion in Section 7.
>
> **[Q3]** We agree with the reviewer that different incorrect classifications may have different consequences—a cold is much more similar to a flu rather than lung cancer in terms of symptoms and treatment. In that context, we acknowledge that, for simplicity, our definition of counterfactual harm treats all incorrect classifications as equally harmful. Expanding our definition of harm to weigh different incorrect classifications according to their consequences would be a very interesting avenue for future work, however, it may be challenging to quantify/measure the consequences. We will discuss this in Section 7 under Assumptions.
>
> **[Q4]** We will expand the discussion in lines 127-130 to include the suggested sentence.

---

> ### Comment · Reviewer_KmhA · 2024-08-12
>
> Dear authors,
>
> Thanks for your detailed response. You have addressed some of my concerns, and I will just share some follow-ups:
>
> (W1) Although I am familiar with the textbook you mention, I was not aware of the do() notation being used for counterfactuals. It does seem that your definition is then correct, I am a happy for you to keep the notation you prefer (I thought there was a genuine mistake). However, two things to keep in mind: (1) I am not sure if there is any place where you mention where your notation is coming from, which would be a very good thing to add. (2) My comment about the notation is still pertinent, in the sense that readers familiar with the framework of the Causality book (Pearl, 2000) will be confused by the notation (this book has 27,000 citations on Google Scholar, if you consider that a relevant metric). For this reason, it may be helpful to have a sentence in the preliminaries that mentions the difference in the notation.
>
> (Q2)
>
> > We would like to clarify that our goal is not to minimize counterfactual harm but rather to find the largest harm-controlling set ...
>
> I am aware of the scope of the paper. However, given that you are spending significant energy on studying this topic, and are cleary doing some good work, I would expect a better justification for how this effort fits into the bigger picture. In other words, I would expect some simple example/illustration to demonstrate how harm-controlling sets you discuss help solve a problem that is not handled by other existing methods. I hope you can share some thoughts, since I would be quite interested to learn about this.

---

> > ### Author Response · Authors · 2024-08-12
> >
> > We would like to thank the reviewer for their follow-up message.
> >
> > **[(W1)]** Following the reviewer's advice and to avoid misunderstandings, we will explicitly point out that we follow Peters et al.'s notation and will also discuss the differences between Peters et al.'s notation and Pearl's notation in the revised version of the paper.
> >
> > **[(Q2)]** After re-reading the original concern raised by the reviewer under Q2, our rebuttal, and the reviewer's follow-up message, we are unsure we understand how to answer to "I would expect some simple example/illustration to demonstrate how harm-controlling sets you discuss help solve a problem that is not handled by other existing methods". Nevertheless, in what follows, we try our best and, if we miss the reviewer's point, please, follow-up further.
> >
> > We view harm control as a prerequisite, not an end goal, for any decision support system in high-stakes domains. In this context, we get inspiration from the medical domain where, before measuring the effectiveness of a new treatment, one measures the harm caused by the treatment. In this context, it is also worth pointing out that, in contrast to our work, existing methods do not leverage the specific monotonicity assumptions of decision support systems based on prediction sets to (partially) identify counterfactual harm directly from observational data.
> >
> > Further, in the case of decision support systems based on prediction sets, we would also like to clarify that the counterfactual and the interventional monotonicity assumptions do not allow us to identify and partially identify counterfactual benefit. In this context, it would be interesting to identify properties under which counterfactual benefit is (partially) identifiable.

---

### Official Review · Reviewer_8Whm · 2024-07-09

**Soundness:** 3
**Presentation:** 3
**Contribution:** 2
**Rating:** 5
**Confidence:** 4

**Summary:**

This paper addresses the issue with prediction methods that generate a set of potential labels, from which a human makes the final decision. The authors identify a harmful scenario where the model's intervention leads to incorrect predictions that a human would have otherwise made correctly. To model this, they use the Structural Causal Model (SCM) framework.

The prediction set size is controlled by a parameter $\lambda$. For a given $\lambda$, they define counterfactual harm as the average number of instances where a human's unaided prediction would be correct, but becomes incorrect with the model's assistance. The main objective of the paper is to identify the range of $\lambda$ values that keep the counterfactual harm within a user-specified constant $\alpha$.

To achieve this, the authors use conformal prediction and propose two training algorithms under two conditions:

- Counterfactual monotonicity: At the instance level $(x, y, \hat{y})$, smaller and correct prediction sets lead to better human predictions.
- Interventional monotonicity: At the sub-population level characterized by $(x, y)$, smaller and correct prediction sets lead to better human predictions *on average*.

They conduct experiments on two image datasets where human predictions from prediction sets are available, demonstrating that their methods effectively control counterfactual harm.

**Strengths:**

The problem formulation is interesting. The authors establish an objective that essentially tunes the threshold parameters used in Conformal Prediction, aiming to control the number of instances where the prediction sets impede the human from making the correct prediction. Solving the optimization problem requires access to counterfactual labels for human predictions across all possible prediction sets, which are naturally unavailable in observational datasets. To address this, the authors identify two weaker conditions, which are highly likely to hold in practice, to solve the problem.

Finally, they apply the conformal prediction algorithm on a hold-out calibration dataset to identify the set of $\lambda$ values that bring the CF harm under control.

**Weaknesses:**

Please refer the questions below.

**Questions:**

**Equation 4:**

In the definition of CF harm, why is the true label $Y$ a function of the predicted label $\hat{Y}$? Could you please elaborate on the notation?

**Line 164:**

Should $v$ not be abducted from the joint distribution of $(X, Y, \hat{Y})$ rather than just $X$?

**Lines 127 to 130:**

I do not understand the text in these lines. Are you stating that one can always find some distribution over the exogenous variables that aligns with the observational data, or are you claiming that the original SCM is identifiable?

**Typo at Line 159:**

has --> had

**Proof for Proposition 1:**

In the proof for Proposition 1, where have you utilized the CF monotonicity condition? The proof seems to hold for any arbitrary SCM $\mathcal{M}$. Am I missing something here?


**On CF monotonicity:**

I believe that under no intervention, a human predicts the label with $C_\lambda(x) = \mathcal{Y}$, i.e., the entire label set. If CF monotonicity holds, why wouldn't any conformal prediction algorithm that guarantees $1-\alpha$ coverage satisfy the CF harm requirement?

**Examples for CF monotonicity violations**

Can you provide examples from your considered datasets where CF monotonicity is violated. In practice, why would such violations happen?

**Tradeoff between CF harm and conformal accuracy:**

I believe accounting for CF harm may unnecessarily increase the prediction set size, beyond what vanilla conformal prediction would offer, for the same amount of coverage guarantees provided by both methods. Can the authors comment on this, at least empirically?

If such a tradeoff exists, this approach would be interesting if one could find the subset of instances where harm occurs and explicitly find the prediction set for such instances. However, I am disappointed that the paper does not propose such approaches.

**Conformal baselines**

The authors should compare their algorithms with the state-of-the-art conformal prediction algorithms and evaluate how their methods perform in terms of (a) conformal accuracy, (b) prediction set size, and (c) CF harm. The current experimental section appears too weak.


**Conformalizing $\hat{Y}$:**

Why shouldn't this approach work? Suppose we apply conformal prediction to both the ground truth $Y$ and the expert prediction $\hat{Y}$, and then emit the prediction set as a union of the two prediction sets obtained from these conformalized models. How does your approach compare to such a proposal in terms of the accuracy $A(\lambda)$ and the prediction set size?

**Limitations:**

Yes, limitations are addressed.

---

> ### Author Rebuttal · Authors · 2024-08-05
>
> **[Eq. 4]** In Eq. 4, the true label $Y$ is not a function of the predicted label $\hat{Y}$. The expectation is over the predicted label $\hat{Y}$ and, following standard notation used elsewhere, the distribution $P^{\mathcal{M} ; do(\Lambda = \lambda) | \hat{Y} = \hat{y}, X = x, Y=y}(\hat{Y})$ denotes the counterfactual distribution entailed by the structural causal model $\mathcal{M}$ under the intervention $do(\Lambda = \lambda)$ given the observation $\hat{Y} = \hat{y}, X = x, Y=y$.
>
> **[Line 164]** The distribution of the noise $v$ is conditioned (only) on $X=x$ because, in Eq. 6, the variable $X=x$ is fixed but the variables $Y$ and $\hat{Y}$ are not fixed.
>
> **[Lines 127-130]** We state the former.
>
> **[Proof of Proposition 1]** We used the CF monotonicity assumption in the 5th line of the proof where we used that $\Bbb{1}\\{\hat{Y} = y\\} = 1$ if $y \in \mathcal{C}\_{\lambda}(x)$. We will clarify this step in the revised version of the paper.
>
> **[On CF monotonicity]** Under the CF monotonicity, any conformal predictor that guarantees $1-\alpha$ coverage is guaranteed to cause average counterfactual harm smaller or equal than $\alpha$. However, there may be conformal predictors that guarantee $1-\alpha$ coverage and cause average counterfactual harm smaller than a strictly smaller bound $\alpha' < \alpha$. This is because the average counterfactual harm does not only depend on the coverage probability but also the probability that a human predicts the ground-truth label successfully on their own.
>
> **[Examples for CF monotonicity violations]** In general, since counterfactuals lie within level three in the “ladder of causation” [19], it is not possible to identify CF monotonicity violations directly from observational data. A violation of counterfactual monotonicity may happen in the following situation. Let $\mathcal{S} \subseteq \mathcal{T}$ be two prediction sets and assume both sets contain the true label $y$. If there is a label value $y' \in \mathcal{T} \backslash \mathcal{S}$ that help the human expert realize that $y$ is the true label, the expert may succeed at predicting $y$ in $\mathcal{T}$ but may fail at prediction $y$ in $\mathcal{S}$.
>
> **[Tradeoff between CF harm and conformal accuracy]** We would like to clarify that, if one sets the value of the threshold $\lambda$ to be roughly the $1-\alpha$ quantile of the empirical distribution of the scores $1-m_{y}(x)$ in the calibration set, then, the set valued predictor defined by Eq. 1 is equivalent to a vanilla conformal predictor with nonconformity scores $1-m_{y}(x)$ and coverage $1-\alpha$. This type of conformal predictors are the state of the art for decision support systems based on prediction sets [17, 18]. Under this view, it becomes apparent that, given the same amount of coverage guarantee, there is not a trade-off between prediction set size and average counterfactual harm and, by searching for $\lambda$ values in $[0,1]$ that are harm controlling, we are essentially searching for vanilla conformal predictors that are harm controlling.
>
> **[Conformal baselines]** As discussed in our previous answer, the class of set valued predictors defined by Eq. 1 are equivalent to the class of vanilla conformal predictors with nonconformity scores $1-m_{y}(x)$.  Moreover, since our focus is on (counterfactual harm of) decision support, we have used the average accuracy of the predictions made by humans using prediction sets constructed by the above set-valued predictors as evaluation metric, similarly as the state of the art [17,18]. However, following the reviewer's suggestion, we have additionally evaluated all the set-valued predictors considered in our experiments in terms of empirical coverage (conformal accuracy) and prediction set size against average counterfactual harm. In the pdf attached with the rebuttal, we summarize the results for the set-valued predictors used in Figure 2.
>
> **[Conformalizing $\hat{Y}$]** Given a set of set-valued predictors and a user-specified bound $\alpha$, the goal of our work is to identify the subset of them that cause average counterfactual harm smaller than $\alpha$. To this end, we have considered the set of set-valued predictors defined by Eq. 1, which as discussed above, are equivalent to the set of vanilla conformal predictors with nonconformity scores $1-m_{y}(x)$ and are the state of the art for decision support systems based on prediction sets [17, 18].
>
> Consequently, given the set of set-valued predictors suggested by the reviewer, or other conformal predictors with different nonconformity scores, one could think of extending our framework to identify the subset of these predictors that cause average counterfactual harm smaller than $\alpha$, however, we leave this as future work. We will clarify this in Section 7 under Methodology.

---

> > ### Comment · Reviewer_8Whm · 2024-08-11
> > **Follow up questions.**
> >
> > Thanks for the rebuttal. I have carefully read your responses. Please see below for the follow up questions.
> >
> > 1. I am particularly interested in understanding the notation $Y=y(\hat{Y})$. Why is the true label expressed as a function of the predicted label?
> >
> > 2. While I understand that $\hat{Y}$ is not fixed. but why is $Y$ not fixed?
> >
> > 3. If conformal predictors with $1-\alpha$ coverage guarantee gives us an $\alpha$ guarantee on CF harm already, wouldn't conformal predictors directly serve as baselines for your approach? In that case, comparing existing SOTA conformal predictors with your method in terms of prediction set size becomes crucial. I’m disappointed that such comparisons were not included in the rebuttal, despite being explicitly asked for.
> >
> > 3. I understand that in observational datasets, it’s not possible to detect CF monotonicity violations. However, in your benchmark datasets, where predictions were modeled based on prediction sets using MNL, did you encounter instances of such violations? Identifying such examples would strongly support the practical motivation for your problem setting. Currently, I am struggling to convince myself that CF harm occurs in practice.
> >
> > 4. I am unclear about the results provided in the PDF. The experiment I requested was as follows: Consider the SOTA CP algorithms in the literature and evaluate the following metrics—CF harm, conformal accuracy, and prediction set size. How does your conformal approach, which explicitly addresses CF harm through the formula $1 - m_y (x)$, compare to other SOTA methods that do not? I’m particularly interested in understanding the gap your approach fills empirically in the existing SOTA methods.

---

> ### Author Response · Authors · 2024-08-12
> **Re: Follow up questions.**
>
> We would like to thank the reviewer for their follow-up message. Below, we provide a point-by-point reply. Please, do not hesitate to follow up further if something is unclear.
>
> 1. In Eq. 4, the true label is not a function of the predicted label. Here, note that $(\hat{Y})$ is not part of the superscript. The expression refers to the probability of the random variable $\hat{Y}$ under the distribution $P^{\mathcal{M}; do(\Lambda=\lambda) | \hat{Y} = \hat{y}, X = x, Y=y}$.
>
> 2. In the definition of CF monotonicity we follow from Straitouri et al. [18], for a fixed $X=x$, the inequality in Eq. 6 needs to hold for any $u$ and $v$ such that $X=x$. However, for different $v$’s such that $X=x$, one may have different $Y$ values since, in the SCM in Eq. 3, we do not make any assumption on $f_X$ and $f_Y$.
>
> 3. and 5. Given a conformal predictor with coverage $1-\alpha$, our framework can be used to predict whether the conformal predictor causes average counterfactual harm smaller than a user-specified bound $\alpha' \in [0, 1]$, with $\alpha' \leq \alpha$. In our experiments, we applied our framework to the SOTA conformal predictors by Straitouri et al. [17,18]. Therefore, in Figure 2 in the one page pdf attached with the rebuttal, we show the (a) average set size and (b) coverage vs the average counterfactual harm achieved by the SOTA conformal predictors by Straitouri et al. [17, 18]. Given this clarification, we are not sure what it would mean to compare these SOTA conformal predictors with our method in terms of prediction set size.
>
> 4. In the benchmark datasets we used in our experiments, we cannot identify counterfactual monotonicity violations using the mixture of MNLs because the mixture of MNL is not a causal model—it is just a prediction model used to estimate the accuracy of the predictions made by humans using prediction sets. However, we would further like to clarify that counterfactual harm may occur even if there are no violations of counterfactual monotonicity. If there are no violations of counterfactual monotonicity, the average counterfactual harm is given by Eq. 8. Therein, it is apparent that, as long as there are samples for which a human expert successfully predicts the true label on their own but the corresponding prediction sets do not include the true label, counterfactual harm will occur. Further, if there are violations of counterfactual monotonicity, the right-hand side of Eq. 8 is a lower bound on the counterfactual harm. Then, based on the results shown in Figures 2 and 3, we can conclude that there exists counterfactual harm irrespective of whether counterfactual monotonicity holds.

---

> > ### Comment · Reviewer_8Whm · 2024-08-13
> > **Thanks for the explanation**
> >
> > 1. Understood, thank you. Apologies for misunderstanding the notation. No further questions.
> >
> > 2. I do not understand this point. You've defined both $X$ and $Y$ solely in terms of the exogenous noise $v$. Why would $X$ vary with $u$? What do you mean by the phrase: "for any $u, v$ such that $X=x$"?
> >
> > I understand that you do not make assumptions about $f_X$ or $f_Y$. So, they can be man y-to-one in general. Let us consider a case where $f_X(v_1) = f_X(v_2) = x$ for $v_1 \ne v_2$, but $f_Y(v_1) = y_1 \ne f_Y(v_2) = y_2$. It seems your definition asserts the following condition:
> >
> > $1\set{f_{\hat{Y}}(u, v_2, C_\lambda(x)) = y_1} \geq 1\set{f_{\hat{Y}}(u, v_2, C_{\lambda'}(x)) = y_1}$
> >
> > Is my interpretation correct? How do you explain the discrepancy between $v_2$ and $y_1$?
> >
> > 3. I disagree with the authors that only the method by Straitouri et al. represents the current state-of-the-art baselines. They should compare their approach with other well-established, such as APS, RAPS, and other advanced conformal predictors. Figure 2 in the provided PDF does not show comparisons with existing leading conformal methods. Such comparisons are crucial to establish the merits of your approach in context of the existing literature.
> >
> > 4. I agree with the authors that CF harm may occur in cases when "a human expert successfully predicts the true label independently, but the corresponding prediction sets do not include the true label." However, many conformal predictors precisely aim to address this issue by ensuring an $\alpha$ coverage guarantee. Such harm is already addressed in my opinion.
> > My main concern has been the following: Suppose we consider the RAPS conformal predictor (for example), which provides an $\alpha$ coverage with CF harm $\alpha' < \alpha$ and has an average prediction set size of $\delta$. If I were to use your Conformal algorithm instead, aiming for the same CF harm $\alpha'$, what would the corresponding prediction set size, and the accuracy be?
> >
> > I am hesitant to recommend an accept, and I retain my score.

---

> > > ### Author Response · Authors · 2024-08-13
> > > **Re: Thanks for the explanation**
> > >
> > > We would like to thank the reviewer for their follow-up message.
> > >
> > > 2. We meant for any $v$ such as $X=x$. However, note that, in the inequality in Eq. 6, the random variable $Y$ is given $f_{Y}(v)$ and thus, in the example brought up by the reviewer, for $v_2$, one has $Y = y_2$. Then, for $u$ and $v_2$, the inequality in Eq. 6 is $\Bbb{1}\\{ f_{\hat{Y}}(u, v_2, C_\lambda(x)) = y_2 \\} \geq \Bbb{1}\\{ f_{\hat{Y}}(u, v_2, C_{\lambda'}(x)) = y_2\\}$ and, for $u$ and $v_1$, the inequality in Eq. 6 is $\Bbb{1}\\{ f_{\hat{Y}}(u, v_1, C_\lambda(x)) = y_1 \\} \geq \Bbb{1}\\{ f_{\hat{Y}}(u, v_1, C_{\lambda'}(x)) = y_1\\}$.
> > >
> > > 3. The conformal predictor used by Straitouri et al. is the only one that has been empirically validated using the same decision support paradigm we use in a large-scale human subject study. Nevertheless, following the reviewer's suggestion, we will apply our framework to APS and RAPS in an Appendix in the revised version of the paper.
> > >
> > > 4. We would like to first clarify that the $\alpha$ coverage guarantee does not take into account whether the human expert successfully predicts the true label independently, and that is the reason why a conformal predictor with $\alpha$ coverage may cause CF harm $\alpha' < \alpha$. Further, in the example brought up with the reviewer, our framework can be used to verify that the RAPS conformal predictor causes CF harm smaller or equal than $\alpha'$. To this end, one needs to use the nonconformity score of RAPS to construct the prediction sets in Eq. 1 rather than the nonconformity score $1-m_{y}(x)$. In doing so, note that our framework does not change the way in which the RAPS conformal predictor constructs prediction sets and thus the prediction set size and accuracy do not change. As discussed under point 3, following the reviewer's suggestion, we will apply our framework to APS and RAPS in an Appendix in the revised version of the paper.
> > >
> > >
> > > Based on the above clarifications and promised changes, we hope you are willing to reconsider your score.

---

> > > > ### Comment · Reviewer_8Whm · 2024-08-14
> > > >
> > > > 2. But you said in your previous response that you abduct $v$ only depending on the $X$. Your definition in Eq. 6 says for any $v \sim P(V|X)$. In my example, both $v_1, v_2$ serve as valid abductions for the $x$. For the above explanation to hold you should have had the definition as for any $v \sim P(V|X, Y)$. But in one of my previous responses you had indicated that abducting with *just* the covariates is the correct thing to do. Your text in the paper and explanation provided above seems disconnected.
> > > >
> > > > 3. I used APS and RAPS as examples, but I assume the authors are familiar with other state-of-the-art conformal predictors. To validate the merits of your conformal formula, it must be compared directly against these leading conformal predictors. I see comparing your approach against the state-of-the-art as the only way to establish its effectiveness.
> > > >
> > > > 4. The $\alpha$ coverage does ensure that the correct label is included in the prediction set at least $1 - \alpha$ of the time. In the absence of strong evidence, it is reasonable to conjecture that CF monotonicity holds. Therefore, all existing conformal predictors indirectly address CF harm and serve as benchmarks for your work.

---

> > > > > ### Author Response · Authors · 2024-08-14
> > > > >
> > > > > 2. Yes, in the definition both $v_1$ and $v_2$ are valid abductions. For each of the abductions, we need Eq. 6 to hold, once for $u, v_1, x$ and $Y = y_1 = f_y(v_1)$ and once for $u,v_2,x$ and $Y=y_2=f_y(v_2)$. In the definition, we fix X=x and we set the value of $Y$ using the mechanism $f_y(v)$ with $v \sim P(v | X=x)$.
> > > > >
> > > > > 3. We would like to once again emphasize that our methodology *does not* propose a way to construct prediction sets or, as the reviewer says, a conformal formula. *Given* a set of set-valued predictors (e.g. a type of conformal predictors), our methodology  can be used to identify which of these predictors cause counterfactual harm less frequently than a user-specified bound.  After applying our methodology, one can use the set-valued predictors off-the-shelf. In our previous reply, we committed to apply our methodology to APS and RAPS but not to compare against them since our methodology is incomparable. If the reviewer feels that we should apply to other specific types of conformal predictors, we would be happy to do so in the revised version of the paper. However, we would like to emphasize once more that such additional experiments will showcase our methodology with other conformal predictors, it *will not* compare our methodology against them.
> > > > >
> > > > > 4. Under the CF monotonicity assumption, a conformal predictor with coverage $1-\alpha$ will cause average counterfactual harm $\alpha$ **if and only if** the human expert succeeds in predicting the true label on their own for any of the $\alpha$ samples for which the true label is not in the prediction set. Note that such a condition *is not* CF monotonicity, it can be actually validated empirically, and it does not hold in practice for any of the classifiers and human predictions we have experimented with. If a human expert fails in predicting the true label on their own for any of the $\alpha$ samples for which the true label is not in the prediction set, then the average counterfactual harm of the conformal predictor will be less than $\alpha$.

---

### Official Review · Reviewer_DuUW · 2024-07-12

**Soundness:** 3
**Presentation:** 2
**Contribution:** 3
**Rating:** 6
**Confidence:** 2

**Summary:**

This paper focuses on decision support designed to help humans in multi-class classification tasks via prediction sets. The paper defines the notion of counterfactual harm, namely when usage the system would lead the agent to a wrong prediction which would have been correct without the system’s ‘help’. The authors show that under some monotonicity assumptions such counterfactual harm is identifiable or partially identifiable. Against this theoretical backdrop, the paper proposes a method to find sets of interventions (namely thresholds for the prediction sets) which are guaranteed to have bounded harm. This approach is validated using experiments on human subject studies. These studies reveal a trade off between post-intervention accuracy and counterfactual harm.

**Strengths:**

I am not an expert in prediction sets so I cannot fully evaluate how significant this contribution is, and was not able to fully check the proofs. I saw the following strengths:
1. The trade off between accuracy and counterfactual harm is very interesting, and the possibility to control the desired amount of counterfactual harm seems very important.
2. The theoretical results seem solid and are well rounded out by the corroborating experiments.
3. Despite the technicality of the paper, I could follow the argument and appreciate the message. The text is well written.

**Weaknesses:**

1.      The notation little y-hat and capital Y-hat in section 3 is somewhat confusing. Line 123 implicitly suggests the latter is the random variable generated by the user predicting with the decision support. Line 142 however states that little y-hat is the prediction the experts would make on their own. I would have expected this to be just   Y-hat but with lambda = 1, as stated elsewhere in the text?
2. In the first experiment, when w is different from 110, the data contains predictions made by humans on their own, and the accuracy of the humans using prediction sets is only estimated. How good are these estimates? We only get to see a comparison with the real value in Figure 3 for w=110, and it is clear that these estimates decisively overshoot.
3. The results show that the sets of lambdas do not contain all the harm-controlling values - as the authors remark - and especially seem to contain less of the ‘good’ values, meaning those that (at the cost of some counterfactual harm within the accepted bound) achieve higher accuracy. This seems to be a serious drawback.
4. Lambda is used in line 105 but only introduced in line 109.
5. Theorem 1 and 2 could use a bit more intuition and/or explanation.

Typos:
line 137: use -> used

**Questions:**

Beside remarks on the weaknesses mentioned above, I would be interested in answers to the following questions:
1.   In Figure 1 you show the average accuracy of humans estimated by MNLs when w=110, but in the text you report that the second dataset contains the predictions of humans with prediction sets at all thresholds for that value of w, which is then shown in Figure 2. Why not start with the actual accuracy then, instead of the estimates? It seems to me that Figure 2 has the core findings, while Figure 1 is more of a sensitivity analysis on the fact that this phenomenon is consistent across model architectures (under the assumptions that the MNL estimates are qualitatively reliable).
2. Three of the models tested have worse average performance than the human experts. What is the point of testing decision support with such models? I suppose we would never implement a decision support that is worse than humans alone (unless perhaps we have strong evidence it is better than humans on a subgroup of data points, but this is not the case here), since it can stir the humans with poor predictions. I can see this could be interesting since you are studying counterfactual harm, but still it is an unrealistic scenario so I would not present results for three such models.

**Limitations:**

The paper seems to mention and discuss all the relevant assumptions and limitations.

---

> ### Author Rebuttal · Authors · 2024-08-05
>
> **[y-hat & Y-hat]** The random variable $\hat{Y}$ always denotes the expert prediction from a prediction set $\mathcal{C}\_{\Lambda}(X)$ following the SCM specified in Eq.3. Under no intervention, $\Lambda=1$ and thus $\mathcal{C}_{\Lambda}(X) = \mathcal{Y}$, as noted in Line 131-132, i.e., the human expert predicts on their own. The variable $\hat{y}$ is always a realization of the random variable $\hat{Y}$, as clarified in Page 3, Footnote 2. Consequently, in Line 142, the variable $\hat{y}$ is the realization of the random variable $\hat{Y}$ under no intervention, i.e., $\Lambda = 1$.
>
>
> **[Estimated accuracy]** To compare the average accuracy estimated using the mixture of MNLs to the average accuracy estimated using predictions made by human participants, we resort to the dataset made publicly available by Straitouri et al. [18]. Unfortunately, this dataset only comprises human predictions for $\omega = 110$ and a single classifier. However, since the reported cost for collecting such data was 7,150 British pounds [18, Section 6], we could not afford gathering additional data to assess the quality of the average accuracy estimated using the mixture of MNLs for other classifiers and $\omega$ values.
>
> While it is clear that the average accuracy estimated using the mixture of MNLs is higher than the average accuracy estimated using predictions made by human participants, as acknowledged in Line 303-306, the estimated accuracy follows the same trend and supports the same qualitative conclusions.
>
>
> **[‘Good’ values in $\Lambda(\alpha)$]** We would like to highlight that, in Figures 2 and 3, the average counterfactual harm $H(\lambda)$ is estimated using data from $50$ random realizations of **a test set**. Here, for each realization of the test set, we apply Theorem 1 and Corollary 1, and include a $\lambda$ value in the set $\Lambda(\alpha)$ if the empirical estimate $\hat{H}_n(\lambda)$ of the average counterfactual harm using data from **a calibration set**, disjoint from the test set, is smaller than the user-specified bound $\alpha$.
>
> Consequently, the closer $H(\lambda)$ is to $\alpha$ from below, the higher the probability that the empirical estimate $\hat{H}_n(\lambda) > \alpha$ due to estimation errors and thus $\lambda \notin \Lambda(\alpha)$. In Figures 2(a) and 3(a), for the particular choice of $\alpha = 0.01$, it turns out the harm-controlling $\lambda$ values with the highest accuracy $A(\lambda)$ (the 'good' values) are those for which $H(\lambda)$ is closer to $\alpha$ and thus are less often included in $\Lambda(\alpha)$. However, this is an inherent limitation of using conformal risk control and, more broadly, distribution-free uncertainty quantification, not a specific drawback of our framework.
>
> That said, during the rebuttal period, we have conducted additional experiments with larger calibration sets. The results are summarized in Figure 1 in the one page pdf attached with the author rebuttal and show that, for larger calibration sets, the ‘good’ values of $\lambda$ are more often included in the set $\Lambda(\alpha)$ since the estimation error is lower.
>
> **[Line 105]** There is a typo in line 105; $\mathcal{C}_{\lambda}(x)$ should have been $\mathcal{C}(x)$. We will fix the typo in the revised version of the paper.
>
> **[Theorem 1 and 2]** We will present an intuitive explanation of Theorem 1 and 2 in the revised version of our paper.
>
> **[Figure 1 and 2]** We agree with the reviewer and will first present the results currently shown in Figure 2 and then the results in Figure 1 in the revised version of the paper.
>
> **[Classifiers worse than human experts]** We would like to point out that, in contrast with decision support systems that provide single label predictions, decision support systems that provide prediction sets such as ours can help humans improve their predictions even if the classifier used to construct the prediction sets has lower average accuracy than the humans, as (empirically) shown by Straitouri et al., ICML 2023 [17; Tables 1 and 2, Figure 3]. In fact, note that, for the three classifiers with worse average performance than the human experts, there always exist values of $\lambda$ for which the average accuracy of the predictions made by humans using our system, as estimated using the mixture of MNLs, is greater than the average accuracy of the predictions made by humans on their own, as estimated using real human data.

---

> > ### Comment · Reviewer_DuUW · 2024-08-08
> > **Reaction to rebuttal**
> >
> > Thanks to the authors for their rebuttal. I believe my points have been addressed sufficiently, and I appreciate the additional work in showing the role of the validation set. I suggest to work their responses of their points 3 and 7 into the main text.
> >
> > Concerning the point of another reviewer on the discussion of the relationship between prediction performance and harm in decision support systems, alongside Wang 2024 and Liu 2024 it would be appropriate to discuss van Amsterdam 2023 at al "When accurate prediction models yield harmful self-fulfilling prophecies", which contains formal results on the topic.

---

> > > ### Author Response · Authors · 2024-08-08
> > > **Re: Reaction to rebuttal**
> > >
> > > We would like to thank the reviewer for their follow-up message. In the revised version of the paper, we will include our responses to points 3 and 7 into the main text and we will discuss the related work suggested by the reviewer alongside Wang et al. (2024) and Liu et al. (2024).

---

### Official Review · Reviewer_RCNh · 2024-07-17

**Soundness:** 3
**Presentation:** 3
**Contribution:** 2
**Rating:** 5
**Confidence:** 4

**Summary:**

This paper relates to decision support systems based on prediction sets. It introduces a framework using conformal risk control to design prediction sets that balance accuracy with minimizing potential harm (understood as a human who has succeeded at predicting the ground-truth label of an instance on their own failing had they used the DS system), validated with human prediction data from empirical lab studies.

**Strengths:**

- Originality. The paper establishes formal definitions/characterizations and a theoretical foundation. The paper formally characterizes the notion of "counterfactual harm" and introduces the counterfactual monotonicity assumptions and weaker interventional monotonicity assumptions to make counterfactual harm identifiable and partially identifiable, respectively. This conceptual contribution improves the understanding of the risks associated with decision support systems based on prediction sets.

- Significance. The authors demonstrated that the idea of "counterfactual harm" can be made practically useful through conformal risk control (section 5), and they performed lab experiments to validate their theoretical bounds. The result is practical: Cor 1 and 2 ensure that these systems cause less counterfactual harm on average than a user-specified threshold.

**Weaknesses:**

1. The main weakness is the usability and motivation of this particular formulation. The paper proposes decision support systems based on prediction sets, which might not always be feasible or realistic, especially in complex domains like healthcare. Most current systems are based on risk prediction e.g. disease risk, drop out risk, for binary outcomes. I don't see any good examples of problems where prediction sets are the right approach, and this paper also does not do this work of illustrating or justifying this formulation using real world case studies.

2. Discussing the Limitations of Predictive Accuracy. While the paper discusses improving accuracy through prediction sets, there may be fundamental limitations in predicting outcomes accurately in the first place. Have the authors considered recent research or critiques regarding the actionability of predictive models? For instance, how does the work align with concerns raised by Wang et al (2024) "Against Predictive Optimization" and Liu et al. (2024) "On the actionability of..."?

**Questions:**

1. Are there practical examples of decision support systems based on prediction sets being successfully implemented in healthcare or other social domains? How do these systems handle the inherent complexity and variability of real-world decision-making scenarios, such as patient care pathways or policy decisions?

2. Have the authors considered recent research or critiques regarding the actionability of predictive models? For instance, how does the work align with concerns raised by Wang et al (2024) "Against Predictive Optimization" and Liu et al. (2024) "On the actionability of..."?

**Limitations:**

Yes.

---

> ### Author Rebuttal · Authors · 2024-08-05
>
> **[Practical examples]** We would like to first clarify that, in the medical domain, prediction sets are often referred to as differential diagnoses. A prominent example of a decision support system that uses patient history and skin condition images to provide differential diagnoses (prediction sets) for skin diseases has been recently developed by Google (see references [1, 2]). This system has been CE-marked as a Class I medical device in the European Union and a study by Jain et al. [3] has concluded that the assistance of this tool was associated with improved diagnoses by physicians and nurses for 1 in every 8 to 10 cases, indicating potential for improving the quality of dermatologic care. We will include this example in the introduction of the revised version of our paper.
>
> [1] Google Health blogpost. Improving access to skin disease information.
>
> [2] Liu, Y., Jain, A., Eng, C. et al. A deep learning system for differential diagnosis of skin diseases. Nature Medicine 26, 900–908 (2020).
>
> [3] Jain, A., et al. Development and assessment of an artificial intelligence--based tool for skin condition diagnosis by primary care physicians and nurse practitioners in teledermatology practices. JAMA network open, 4(4), e217249--e217249 (2021).
>
> **[Complexity and variability of real-world decision-making scenarios]** Decision systems based on prediction sets such as ours and the example highlighted in the previous reply aim to help decision makers make more accurate predictions about outcomes of interest in real-world decision making scenarios. However, more research is needed to evaluate the impact that these systems and their predictions may have on patient care pathways or policy decisions.
>
> **[Limitations of predictive accuracy]** Since Wang et al. (2024) and Liu et al. (2024) were published shortly before the NeurIPS deadline, we were unintentionally unaware of their critique. However, during the rebuttal period, we have carefully read both papers and we think our work actually aligns with the critiques against prediction optimization raised by Wang et al. (2024) and Liu et al. (2024). More specifically, in lines 115-117 of our paper, we argue that improving accuracy may come at the cost of causing harm. Motivated by this observation, the goal of our work is to identify systems that are guaranteed to cause less harm, on average, than a user-specified bound. In Section 7, we discuss other critiques raised by Wang et al. (2024), such as addressing fairness considerations and the challenge of distribution shift. In the revised version of the paper, we will cite and discuss Wang et al. (2024) and Liu et al. (2024) in the introduction and related work.

---

### Author Rebuttal · Authors · 2024-08-05

We would like to thank the reviewers for their careful and insightful comments, which will help improve our paper. Please, find a point-by-point response below and a one page pdf with additional results attached. In the revised version of the paper, we will fix all typos pointed out by the reviewers.

---

### Decision · Program_Chairs · 2024-09-25

**Decision:**

Accept (poster)

**Comment:**

All four reviewers were on the side of acceptance before the author rebuttal and discussion. However, one reviewer became more negative over the course of a long discussion with the authors and myself, although their score remains at borderline accept. I think that the reason for this more negative sentiment, while valid, does not outweigh the strengths of the submission, as I discuss below.

Reviewers found strengths in all aspects of the work. At the conceptual level, reviewers appreciated the formal definition of counterfactual harm caused by a prediction set. The identification of monotonicity assumptions (both a stronger and a weaker one) needed to estimate counterfactual harm is a strength in terms of theory. A method of controlling counterfactual harm is proposed, and the experiments with human subjects were also seen as a strength. Lastly, reviewers found the submission to be clearly written despite the technicality.

The rebuttal satisfactorily addressed several issues in my view, including practical applications of prediction sets, the "quality" of $\lambda$'s determined to be harm-controlling (a limitation of conformal risk control alleviated by a larger calibration set), models having worse performance than humans, confusion over causal inference notation, and an unclear relationship with the work of Richens et al. Clarifications and improvements to the paper are within the scope of a camera-ready revision.

Reviewer 8Whm engaged in a long discussion with the authors and later with myself. While part of this discussion was on more technical points, the most serious concern for this reviewer in the end was the question of comparing with additional state-of-the-art conformal predictors, in terms of counterfactual harm, coverage (conformal accuracy), and prediction set size. I think the discussion clarified somewhat that conformal predictors and the proposed method play complementary roles, where the latter provides more refined estimates of counterfactual harm for the former. Nonetheless, the reviewer wanted to know how the proposed method of choosing $\lambda$ affects conformal accuracy and prediction set size, for additional conformal predictors. I think this is a valid request. However, I do not see it as a core concern for the human-AI setting considered in the paper, where the prediction set is provided as an intermediate quantity for the human to select from, not as a final output, and human accuracy and counterfactual harm appear to be more relevant metrics. Thus, I do not think this concern should hold the submission back.  I do encourage the authors to follow through on their promise of experimenting with additional conformal predictors and to report all metrics, including conformal accuracy and prediction set size.